# Selective PPAR-Delta/PPAR-Gamma Activation Improves Cognition in a Model of Alzheimer’s Disease

**DOI:** 10.3390/cells12081116

**Published:** 2023-04-08

**Authors:** Ian Steinke, Manoj Govindarajulu, Priyanka Das Pinky, Jenna Bloemer, Sieun Yoo, Tracey Ward, Taylor Schaedig, Taylor Young, Fajar Setyo Wibowo, Vishnu Suppiramaniam, Rajesh H. Amin

**Affiliations:** 1Department of Drug Discovery and Development, Auburn University, Auburn, AL 36879, USA; 2Department of Pharmaceutical and Biomedical Sciences, Touro College of Pharmacy, New York, NY 10027, USA; 3Department of Pharmaceutical Sciences, Ferris State University, Big Rapids, MI 49307, USA; 4College of Science and Mathematics, Kennesaw State University, Kennesaw, GA 31044, USA

**Keywords:** peroxisomal proliferator activating receptor, in silico drug design, neurodegeneration, Alzheimer’s disease, synaptic plasticity, behavioral deficits, dendritic spines

## Abstract

*Background:* The continuously increasing association of Alzheimer’s disease (AD) with increased mortality rates indicates an unmet medical need and the critical need for establishing novel molecular targets for therapeutic potential. Agonists for peroxisomal proliferator activating receptors (PPAR) are known to regulate energy in the body and have shown positive effects against Alzheimer’s disease. There are three members of this class (delta, gamma, and alpha), with PPAR-gamma being the most studied, as these pharmaceutical agonists offer promise for AD because they reduce amyloid beta and tau pathologies, display anti-inflammatory properties, and improve cognition. However, they display poor brain bioavailability and are associated with several adverse side effects on human health, thus limiting their clinical application. *Methods*: We have developed a novel series of PPAR-delta and PPAR-gamma agonists in silico with AU9 as our lead compound that displays selective amino acid interactions focused upon avoiding the Tyr-473 epitope in the PPAR-gamma AF2 ligand binding domain. *Results*: This design helps to avoid the unwanted side effects of current PPAR-gamma agonists and improve behavioral deficits and synaptic plasticity while reducing amyloid-beta levels and inflammation in 3xTgAD animals. *Conclusions*: Our innovative in silico design of PPAR-delta/gamma agonists may offer new perspectives for this class of agonists for AD.

## 1. Introduction

Peroxisome proliferator-activated receptors (PPARs) are members of the nuclear hormone receptor superfamily that are ligand-activated transcription factors [1]. These receptors are associated with many systemic and cellular functions, including insulin sensitivity and whole-body energy regulation [2,3]. PPARα is abundantly expressed in tissues that utilize fatty acid catabolism, such as the heart, liver, brown adipose tissue, and the kidney. PPARγ, which exists in two isoforms, γ1 and γ2, is most abundant in expression in adipose tissue and regulates adipocyte differentiation and lipid storage [1]. PPARδ/β has a wide range of expressions and is associated with activity in the skeletal muscle, the gut, and the brain. Interestingly, all three forms are expressed in the brain; however, PPARδ/β is the most abundant in the brain, specifically in neurons and microglial cells [4]. PPARγ is less observed when compared to PPARδ/β in these cells, and PPARα is observed mainly in astrocytes. However, PPARγ agonists are the most extensively investigated form of PPAR agonist for AD therapy. They may serve as potential therapeutic targets for AD because of their positive effects against pathologies and their learning and memory-enhancing effects in transgenic AD animal models [5,6]. Orally administered rosiglitazone (Rosi) at high dosing concentrations (9–18 mg/kg) and extended times (three months) was observed to improve spatial memory and long-term potentiation (LTP) in diabetic and AD rodent models [6,7,8]. To date, most studies involving pharmacologically activated PPARγ have focused on anti-inflammatory mechanisms and thus altered amyloid-beta (Aβ) and Tau pathology levels. For example, pioglitazone (Pio) treatment (18 mg/kg daily) for 5 weeks improved memory in STZ-diabetic mice on high-fat diets by reducing Aβ40/Aβ42 via inhibition of NF-kB, BACE1, and RAGE in the brain, as well as attenuating hyperglycemia [9]. However, mixed results for Pio in mouse AD models make the application of PPARγ agonists for AD questionable [10]. To further dampen the enthusiasm of PPARγ agonists or thiazolidinediones (TZDs) for AD, these classes of drugs display poor blood–brain barrier (BBB) permeability and deleterious effects on human health [5,11]. Many studies utilizing Pio and Rosi require high concentrations of the drug over an extended period of time to obtain therapeutic effects for improving pathologies associated with AD. Unfortunately, higher concentrations of Rosi or Pio treatment lead to unwanted off-target effects that are life-threatening in humans. Further clinical trials investigating the efficacy of Pio with long-term treatment failed due to complaints from patients with increased weight gain and significant edema [12].

These failures to safely ameliorate or mitigate AD development in animal models and at the clinical level have quenched the clinical applicability of these agonists. Further, they have negated volumes of positive findings verifying these therapeutics’ ability to reduce pathology and neurodegeneration associated with AD. Therefore, it is critical to develop novel PPAR-targeted agents that display improved bioavailability and tolerability. Second, because PPARδ is the most abundant PPAR nuclear receptor in the brain, this may offer a new therapeutic target for AD therapy. Recent work in the field of PPAR biology has focused on dual PPAR agonists for AD therapy [13]. The present manuscript discusses the design and development of a potential next-generation PPAR agonist for improving behavioral deficits, synaptic plasticity, and pathologies in a 3xTgAD mouse. We have rationally developed a novel dual PPARδ/γ agonist in silico. Twenty-three nontraditional lead compounds were designed, synthesized, and tested [14]. The design of the PPARγ compounds was based on avoiding the tyrosine-473 site in the AF2 ligand binding domain. Therefore, compounds that displayed robust PPARδ and partial PPARγ activity were then further evaluated for biological significance. The evaluation of these compounds allowed us to advance our lead compound, AU9, for further investigation for improving behavioral deficits, synaptic plasticity and reducing amyloid beta in 3xTgAD mice.

## 2. Materials and Methods

Animals: 3xTgAD mice (B6;129-Tg(APPSwe, tauP301L) and control (C57BL/6) female mice (wild-type) were obtained from The Jackson Laboratory (stock #008195 and #101045, respectively). According to the Jackson Laboratory, the 3xTgAD male mice exhibit fewer phenotypic traits when compared to females, and hence only female mice were utilized in the current study. All mice were group-housed (4 per cage) with free access to food and water in a temperature- and humidity-controlled colony room with a reverse 12:12 light/dark cycle, thus allowing proper timing of animal studies to occur at normal working hours. All experiments and procedures were conducted in accordance with National Institute of Health (NIH) guidelines and approved by Auburn University Institutional Animal Care and Use Committee (IACUC) and following the ARRIVE guidelines.

Drug treatment: AU9 was reconstituted in saline and administered orally (5 mg/kg) daily, starting at 9 months of age and continuing until 12 months of age. AU9 was designed to have higher water solubility and lipophilicity than traditional TZDs. The calculated partition coefficient (oil/water) of AU9 is on a scale from −2 to +2 of 1, where 2 is highest.

Cell lines: BV2: The murine microglial cell line BV-2 was purchased from ACCEGEN (Cat#ABC-TC212S) and cultured according to the manufacturer’s recommendation. Briefly, cultured cells were grown in Dulbecco’s Modified Eagle Medium (DMEM) supplemented with 10% fetal bovine serum and 1% penicillin-streptomycin in a humidified CO_2_ incubator. Chinese Hamster Ovary cells–CHO expressing Swedish mutant APP (APPswe) and wild type human PSEN1, were a gift from Dr. Sasha Waggen [15]. Cells were grown in Dulbecco’s Modified Eagle Medium (DMEM) supplemented with 10% fetal bovine serum and 1% penicillin-streptomycin and grown in a humidified CO_2_ in humidified atmosphere of 5% CO_2_/95% air at 37 °C. The cells were cultured in the presence of G418 (200 μg/mL, Invitrogen) and puromycin (7.5 μg/mL, ThermoFischer Scientific, Waltham, MA, USA) to maintain selection for the expression plasmid. The cells were plated at an appropriate density according to each experimental scale.

Chemicals: AU9 was synthesized by Dr. Tracey Ward at Ferris State University. The drug synthesis scheme and validation of purity have been described previously [14].

Modeling: The Schrodinger software suite was used to perform computational analysis of the interactions of AU9, GW0742 and Pio with their respective PPAR ligand binding domains. PDB crystal structures were used to confirm ligand receptor interactions. Ligand docking studies were performed to determine the most stable docking poses determined by the ligand docking scores, which represent the free energy upon binding of the ligand to the proteins active site. Using the lowest energy conformation, a model system was built to explore the molecular dynamics of this interaction using a simulated annealing technique. For comparison, a full PPARδ agonist, GW0742, was used to illustrate key differences in our compound’s ability to achieve similar transcriptional activity in vitro.

Protein Preparation: Molecular models for PPARβ/δ and γ were built using the ligand conformation obtained from X-ray crystallographic structures of GW0742 bound to PPARδ (PDB: 3TKM) and Rosiglitazone bound to PPARγ (PDB:5Y2O). Protein crystal structures were imported and prepared using the Maestro modeling software protein preparation workflow. In preprocessing of the protein structures, termini were capped and any missing chains were filled in using Prime. H-bond optimization was performed using PROPKA at a pH of 7.4. Lastly, restrained minimization was performed with convergence of heavy atoms to RMSD of 0.30 Å and deletion of all water molecules within 5 Å of the ligand utilizing the force field OPLS4.

Induced Fit Docking: All ligands were prepared using LigPrep with the OPLS4 force field. Ligands were ionized at a pH of 7.4 ± 0.2 using Epik. Prepared ligands were then subjected to induced fit docking by selecting the centroid of the workspace ligand in each protein complex. Residues were refined within 5.0 Å of ligand poses. Glide re-docking was performed using standard precision. The lowest energy docking score for each ligand was evaluated and used for further molecular modeling experiments.

Model System Generation for Molecular Dynamics: Model systems were built from the best induced fit docking poses using a predefined simple point-charge (SPC) water solvent model. An orthorhombic box shape was chosen with a salt concentration of 0.15 M. The model system was built with the force field OPLS4. The method of simulated annealing was used to evaluate molecular dynamics. Each previously built model system was loaded into the simulation from the workspace. Simulation parameters were set to have a schedule of seventeen temperature changes over the course of 1.2 ns using an NVT ensemble class at 1.01325 bar. Model systems were relaxed prior to simulation.

Behavioral Studies: Novel Object Recognition: Object recognition testing was performed as previously described [16,17,18]. Briefly, two days before the training, each mouse was handled gently for 5 min and then allowed to familiarize with the apparatus (a plexiglass box 40 cm × 40 cm and 15 cm high) for 10 min per day. The object recognition test consisted of two 10 min trials, one per day. This extended exposure allowed the animals to learn the task. In the first trial (T1), two identical objects were placed in the central part of the box, equally distant from the perimeter. Each mouse was placed in the apparatus and allowed to explore it. Exploration was defined as the mouse pointing its nose toward the object from a distance of no more than 2 cm (as marked by a reference circle). The mouse was then returned to its cage. The second trial (T2) was performed 24 h later to test memory retention. Mice were presented with two objects, a “familiar” (i.e., the one used for T1) and a “novel” object. The last object was placed on the left or the right side of the box in a randomly but balanced manner to minimize potential biases due to a preference for particular locations or objects. To avoid olfactory cues, the objects and the apparatus were cleaned with 70% ethanol after each trial. Recordings were measured by blind reviewers. Exploration of the objects was defined as time spent with the snout orientated toward the object at a distance of <2 cm of the object. Results were expressed as a discrimination index (DI) (T novel − T familiar)/(T familiar + T novel). The following parameters were evaluated: exploratory object preference and time of exploration of the two objects expressed as % exploration of the familiar and % exploration of the novel object; and discrimination.

Y-maze Test: Spatial recognition memory utilizing a two-trial Y-maze task was performed as previously described [16,19,20,21,22]. Briefly, the plastic Y-maze apparatus consisted of three arms, with each arm separated by 120 degrees, and visual cues were placed around the Y-maze. The two trials were separated by a 3 h inter-trial interval to evaluate spatial recognition memory. During the first trial (acquisition), mice were allowed to freely explore the two arms of the maze for 10 min. For identification, one arm was the starting arm, where the mice were initially placed, and a second arm was identified as the familiar arm; while a third (novel) arm was closed. During the second trial (retention test), mice were placed back in the starting arm and allowed to explore for 6 min with free access to all three arms (the novel arm was opened). To eliminate odors between animals, the entire Y-Maze, including the arena, was cleaned with 70% ethanol. Blinded reviewers scored recordings, and the total number of entries and time spent in each arm were measured. All data are expressed as means ± SEM. Statistical analyses were performed for all behavioral studies using Student *t*-test with Tukey’s post hoc analysis for comparing specific groups. (*p* < 0.05 was considered to indicate statistical significance).

Electrophysiology studies, Hippocampal slice preparation: Animals were euthanized with carbon dioxide, and 350 μm-thick transverse slices were prepared using a Leica VT1200S Vibratome (Leica Microsystems, Wetzlar, Germany). Slices were incubated at room temperature in artificial cerebrospinal fluid (ACSF; 124 mM NaCl, 2.5 mM KCl, 1.5 mM MgCl_2_, 2 mM CaCl_2_, 1.25 mM NaH_2_PO_4_, 25 mM NaHCO_3_, 25 mM dextrose, pH 7.4) saturated with 95% O_2_/5% CO_2_ until transfer to the recording chamber.

Extracellular field potential recording: Brain slices were incubated for at least two hours in ACSF and then transferred into a recording chamber for electrophysiological measurements as previously described with continuous ACSF perfusion at 34 °C [19,23,24]. A bipolar stimulating electrode (MicroProbes, Gaithersburg, MD, USA) was placed in the Schaffer collateral pathway. An extracellular recording pipette drawn with the PC-10 Dual-Stage Glass Micropipette Puller (Narishige, Amityville, NY, USA) and filled with ACSF (2–6 MΩ) was placed in the stratum radiatum of CA1 to record field excitatory postsynaptic potentials (fEPSPs). For LTP experiments, stimulus intensity was set at 50% of the amplitude, at which the preliminary population spike appeared. LTP was then induced after 10 min of stable baseline recording using a Theta Burst Stimulation (TBS) protocol (10 bursts of stimuli, each of four pulses at 100 Hz, interburst interval of 200 ms, and 20 s intervals between individual sweeps), and recording was continued for 60 min post-TBS [19,23,24]. LTP was measured as an average of fEPSP slopes from 50–60 min after the end of induction. The data were recorded online using the WinLTP software (University of Bristol, UK). Standard offline analyses of the data were conducted using Prism software (GraphPad Prism version 8, San Diego, CA, USA).

Western blot analysis: Hippocampi tissue from 3xTgAD and wild type (6 mice per group ± AU9) vehicle-control and drug-treated mice were homogenized in a neuronal lysis buffer (N-PER; Neuronal Protein Extraction Reagent, ThermoFisher Scientific) containing a protease cocktail inhibitor (Halt Protease cocktail inhibitor). Lysate was cleared by centrifugation at 4 °C at 12,000× *g* for 20 min. Cleared lysate was collected and the total protein was estimated by Nanodrop (280/260 wavelength) and stored at −80 °C until use. Lysate was mixed with 4X Laemelli buffer containing DTT and heated at 85 °C for 5 min. Protein homogenate was resolved via a 4–16% SurePAGE precast gel (GenScript Biotech), and transferred to nitrocellulose (BioRad, Hercules, CA, USA) by semi-dry techniques (BioRad). The immuno-blots were blocked with 5% bovine serum albumin in Tris-buffered saline containing 0.1% Tween 20 (TBST) for 1 h, followed by 3x washes with TBST and incubated with primary antibodies overnight and 4 °C. The following day, blots were washed, and probed with secondary anti-rabbit or anti-mouse antibodies (Cell Signaling Technology, Danvers, MA, USA, 1:2000 in TBS-T+BSA solution) for 2 h. Immunoblots were then exposed to ECL reagent (Millipore) and imaged using a LICOR imager. The analyses of bands were based upon densities that were standardized to alpha-tubulin. All data are expressed as means ± SEM. Statistical analyses were performed using Student *t*-test with Tukey’s post hoc analysis for comparing specific groups. (*p* < 0.05 was considered to indicate statistical significance).

Reporter assays: To validate the specificity of the compounds towards the activation of PPARδ and PPARγ, we utilized a PPARδ or PPARγ ligand binding domain driven GAL4 reporter HEK293 stable cell line system, purchased from Signosis (SanDiego, CA, USA). Briefly cells were plated into 24-well plates in triplicate, with 6 independent plates, followed the next day with increasing concentrations of AU9, GW074 or Pio (1 nM–20 µM). Luciferase activity was accomplished using Bright-Glo, assay system (Promega, Madison, WI, USA) and standardized to total protein concentration per well. Alternatively, compounds were tested for the capacity to bind to select PPARδ and PPARγ DNA recognition elements, Peroxisome Proliferator Response Elements (PPRE). The PPRE are unique sites located in the promoter region where PPARs bind and transcriptionally activate the target genes. AP2-PPRE is the PPARγ target involved in adipocyte growth and differentiation and a kind gift from Bruce S. Spiegelman (Addgene) [25]. To test PPARδ activity, we utilized a p4xDRE-Luc plasmid, a kind gift from Bert Vogelsein (Addgene) [26]. These vectors were co-transfected with a Renilla vector (promega) into HEK-293 cells using Jet Prime (PolyPlus, France). Relative light units (RLU) were measured using a Glomax Luminometer (Promega, Madison, WI, USA). Data were standardized to Renilla activity using Dual glo assay (Promega). VP16 vector was used for constitutively active PPARγ and was a kind gift from Mitch Lazar at the University of Pennsylvania. Mutated Human PPAR-gamma Tyr-473 to Phenylalanine plasmid was purchased from Sinobiological. Statistical analyses were performed using Student *t*-test with Tukey’s post hoc analysis for comparing specific groups. (*p* < 0.05 was considered to indicate statistical significance).

Gene expression: RNA was extracted from the hippocampi regions (6 mice per group, with 2 hippocampi per mouse combined together) from mice brains using Trizol (Invitrogen). Approximately 200 ng of RNA was converted to cDNA using OneScript Plus cDNA synthesis Kit (Applied Biological Material) followed by qPCR analysis using BlasTaq 2c qPCR MasterMix. Primers used for qRT-PCR were purchased from IDT. Please see the table of primers in the Appendix A. Data were represented by ΔΔCT based upon gene of interest cycle numbers standardized to beta actin ct values.

Rapid Golgi Staining Procedure: Golgi Cox staining procedure followed a previously published protocol [27]. Briefly, whole brains were harvested from mice and stained using the FD Rapid GolgiStain kit (FD NeuroTechnologies). Brains were immersed in a 1:1 mixture of FD Solution A:B for 2 weeks at room temperature in the dark and then transferred to FD Solution C and kept in the dark for an additional 48 h. Solution C was replaced after the first 24 h. Brains (6 Wt ± AU9 and 3xTgAD ± AU9) from 12-month female mice treated with AU9 for three months daily (5 mg/Kg) were cut into approximately 200 µM sections using a Leica vibratome, with no less than 10 slices from regions containing hippocampi, were transferred to gelatin coated slides onto small drops of FD Solution C and sealed using Permount mounting media (. Ten neurons per slice were imaged using a Nikon Ti inverted microscope from the CA1 and CA3 regions of the hippocampus using a z-stacking procedure with 20 slices per neuron with 0.1 µM per optical slice.

Spine length refers to the sum of the lengths of all spine branches on neurons (µm). The spatial density/volume of a spine was the smallest cubic volume that could image the entire spine (µm^3^). Student *t*-test was used to analyze the differences among groups. Confidence level was set to 0.05 (*p*-value) and all the results are presented as the mean ± SEM.

Neurotrophin measurement: The levels of mouse neurotrophins (NGF, BDNF, NT3, and NT4/5) were measured from mice hippocampi (6 mice per group (wt and 3xTgAD) ± AU9) using a commercial ELISA kit (Biosensis) (Cat# BEK-2231). To measure neurotrophin levels, soluble proteins were extracted using a protocol based upon Kolbeck et al. [28]. Briefly, hippocampi were suspended in 20 volume/weight extraction buffer (0.05 M sodium acetate, 1 M sodium chloride, 1% Triton-X100, Roche complete inhibitor cocktail tablet) and homogenized. Protein concentrations were standardized using Nanodrop, followed by an ELISA for neurotrophins, according to the manufacturer’s instructions (Biosensis). The resulting measurements (pg) were normalized per mg of total soluble protein. Hippocampal homogenate neurotrophin concentrations were based upon a standard curve from the known concentration and measured by a plated reader at 450 nm. Student *t*-test was used to analyze the differences among groups. Confidence level was set to 0.05 (*p*-value) and all the results are presented as the mean ± SEM.

Immunostain for Aβ: Cryosections (3–5 sections per sample, 10–15 µm each, six brains per Wt and 3xTgAD ± AU9) were taken from the CA3–CA1 regions of the hippocampus using a Leica cryostat and fixed with 4% formaldehyde for 10 min followed by permeabilization with 0.1% triton-100 in PBS. Sections were washed three times in PBS, 5 min each and blocked for 2 h in 5% goat serum/PNS solution. Sections were washed three times again in PBS and exposed to primary antibody overnight at 4 °C (1:500 dilution overnight with 5% goat serum overnight). Anti-6E10 antibody (Biolegend) Alexa Fluor 488, which is reactive to aa 1–16 Aβ and to APP, reacts to the abnormally processed isoforms, as well as precursor forms. The following day sections were washed three times in PBS and counterstained with Dapi (Sigma chemical) and mounted with a coverslip using an antifade solution (Molecular probes). Sections were imaged using an inverted fluorescence NikonT1 microscope. The immunofluorescence stained area was determined by the density of immunostain standardized to the total area using ImageJ software. Goat serum was used in place of the primary antibody and was used as a negative control. Images of 3xTgAD with the primary antibody were used as the baseline for time and exposure levels and were then used for all images obtained with these values to nullify background levels.

Aβ ELISA assay in mice: Mouse hippocampi from treated and untreated mice as discussed above (6 per group) Wt and 3xTgAD mice were collected to detect the secreted Aβ1–42 based on the manufacturer’s protocol (R&D Systems). The Aβ1–42 concentrations were quantified using values from a standard curve associated with the ELISA kits following the manufacturer’s protocol. The optical densities of each well were measured at 450 nm using a microplate reader (Agilent, Santa Clara, CA, USA) and the sample Aβ1–42 concentrations were determined by comparison with the Aβ1–42 standard curves. All readings were in the linear range of the assay. Values were standardized to total protein concentrations. Student *t*-test was used to analyze the differences among groups. Confidence level was set to 0.05 (*p*-value) and all the results are presented as the mean ± SEM.

Aβ ELISA assay in APP-Cho cells: AppCho cells were plated in triplicate with six independent plates with increasing concentrations of AU9 (1–20 µM) (24 h). Media was collected the following day to detect the secreted Aβ1–42. The Aβ1–42 concentrations were quantified using values from a standard curve associated with the ELISA kits following the manufacturer’s protocol. The optical densities of each well were measured at 450 nm using a microplate reader (Agilent, Santa Clara, CA, USA) and the sample Aβ1–42 concentrations were determined by comparison with the Aβ1–42 standard curves. All readings were in the linear range of the assay. Values were based upon a known standard curve and standardized to total protein concentration. Student *t*-test was used to analyze the differences among groups. The confidence level was set to 0.05 (*p*-value) and all the results are presented as the mean ± SEM.

β-secretase activity assay: β-site-APP cleaving enzyme (BACE) or β-Secretase activity was determined fluorimetrically using an β-Secretase activity kit (BioVision, Waltham, MA, USA). APP-Cho cells were pleated in triplicate in 24-well plates with 6 independent plates for each experiment and treated with AU9, GW0742 and Pio for 24 h (10 µM). Values were determined based upon manufacturer’s instructions and a standard curve. Beta-secretase activity was represented as relative fluorescence units per mg of total protein. Values were based upon known standard curve and standardized to the total protein concentration. Student *t*-test was used to analyze the differences among groups. Confidence level was set to 0.05 (*p*-value) and all the results are presented as the mean ± SEM.

Nanostring Gene Expression analysis: Appendix A. Hippocampal RNA from 3xTgAD and control mice treated with either saline or AU9 were extracted and purified using an RNA Plus Universal Mini Kit (Cat. #73404, QIAGEN, Germantown, MD, USA). For nCounter analysis, total RNA was diluted to 20 ng/μL and probed using a mouse nCounter Neuropathology Panel (Nanostring Technologies, Seattle, WA, USA). Counts for target genes were normalized to the best-fitting housekeeping genes as determined by nSolver software. The tables observed in the Appendix A are based on units associated with the neuropathology panel. Included are results for Nuerotransmission Appendix A, Cellular Stress (Appendix A), Cytokine and the associated signaling markers (Appendix A), and markers associated with DNA damage (Appendix A). Table for QPCR primers (Appendix A). Western Whole blots for PSD95, GluA1, GluA2, and the associated standardizing Alpha-Actin are shown in Appendix A. Western whole blots for inflammatory markers, including IBA1 and TSPO, are shown in Appendix A and their associated standardizing marker alpha-actin.

Nitrite content: BV2 microglia (2 × 10^5^ cells/mL) were seeded in the 96-well plates as triplicate in each plate and each plate was repeated six times for 12 h, followed by AU9 treatment for 12 h (5–100 µM). Media were changed and LPS (Sigma L2654,100 ng/mL) was added to the media for 24 h. Cultured supernatant was then collected, centrifuged (2500 r.p.m for 20 min) and 100 µL was added to 100 µL of Griess reagent (1% sulfanilamide and 0.1% naphthylethylenediamine dihydrochloride in 2.5% phosphoric acid; Promega G2930, Madison, WI, USA) for 10 min in the dark at room temperature. An ELISA microplate reader was used for the measurement of absorbances at 540 nm. A standard curve was generated in the same manner using NaNO_2_ for quantitation. Student *t*-test was used to analyze the differences among groups. Confidence level was set to 0.05 (*p*-value) and all the results are presented as the mean ± SEM.

Statistical analysis: All data are expressed as means ± SEM. Statistical analyses were performed using a Student *t*-test or a two-tailed, unpaired *t*-test. Additionally, Tukey post- hoc comparisons were used to compare groups when analysis of variance indicated significant effects, except where expected effects were assessed with planned comparisons. In all cases, *p* < 0.05 was considered to indicate statistical significance. All statistical analyses were performed using the GraphPad Prism version 9 software (La Jolla, CA, USA).

## 3. Results

In silico design of AU9: PPARδ Site Map Description. The PPARδ ligand binding domain (LBD) consists of a Y-shaped hydrophobic cavity with three functionally different arms, identified by the computationally derived surface site map seen in Figure 1 (PDB: 3TKM). Arm 1 contains the highly conserved helix 12 (H12) C-terminus, referred to as the activation function 2 (AF2) domain [29,30]. Full agonists have been shown to form strong hydrogen bond interactions in the AF2 LBD contained in arm 1. Further stabilization of the ligand–protein complex is achieved by hydrophobic interactions in arm 2. While the AF2 domain is highly conserved across all PPAR isoforms, functional differences within the ligand binding pocket modulate substrate selectivity [31]. Observable differences in the PPARδ’s site map descriptors can be seen in Figure 1, highlighting a narrow hydrophobic entrance to the AF2 domain with minimal polar contacts colored yellow and red, respectively. The strength of ligand induced activation of the AF2 domain is strictly controlled by access to the tyrosine 437 residue, colored purple in Figure 1B. The supporting polar contacts in arm 1, coming from the histidine 413 residue on H11 and histidine 287 residue on H5, restrict access of sterically hindered hydrophilic groups to tyrosine 437 (Figure 1C,D). Figure 1C demonstrates the interactions of full PPARδ agonist GW0742 bound within the active site. GW0742 has been shown to have a 300- to 1000-fold preference for PPARδ over the other PPAR isoforms [32]. The GW0742 specificity and strength of activation for PPARδ can be observed in the polar phenoxy acetic acid functional group that can extend deep into arm 1 to form a bifurcated hydrogen bond with tyrosine 437 and histidine 413 (Figure 1C). Additional hydrogen bonding to histidine 287 provides further stability and coordination to the AF2 domain for co-activator recruitment. Additionally, the hydrophobic tail of GW0742 takes advantage of arm 2 hydrophobic contacts to valine 305, tryptophan 228, and valine 312, providing stability for the heterodimerization to RXR.

### 3.1. PPAR Delta Induced Fit Docking and Molecular Dynamics

To further probe the interactions that AU9 has with the PPARδ LBD, the PDB crystal structure 3TKM was used, which has the PPARδ LBD with full agonist GW0742 bound. Induced fit ligand docking studies were performed to determine the lowest energy conformation, evaluated by the ligand docking scores, which represent the free energy upon binding of the ligand to the protein active site. While computationally more intensive, induced fit docking accurately accounts for both ligand and receptor flexibility [33]. Using the lowest obtained energy conformation, a model system was built to explore the molecular dynamics of this interaction using a simulated annealing technique.

**Figure 1 cells-12-01116-f001:**
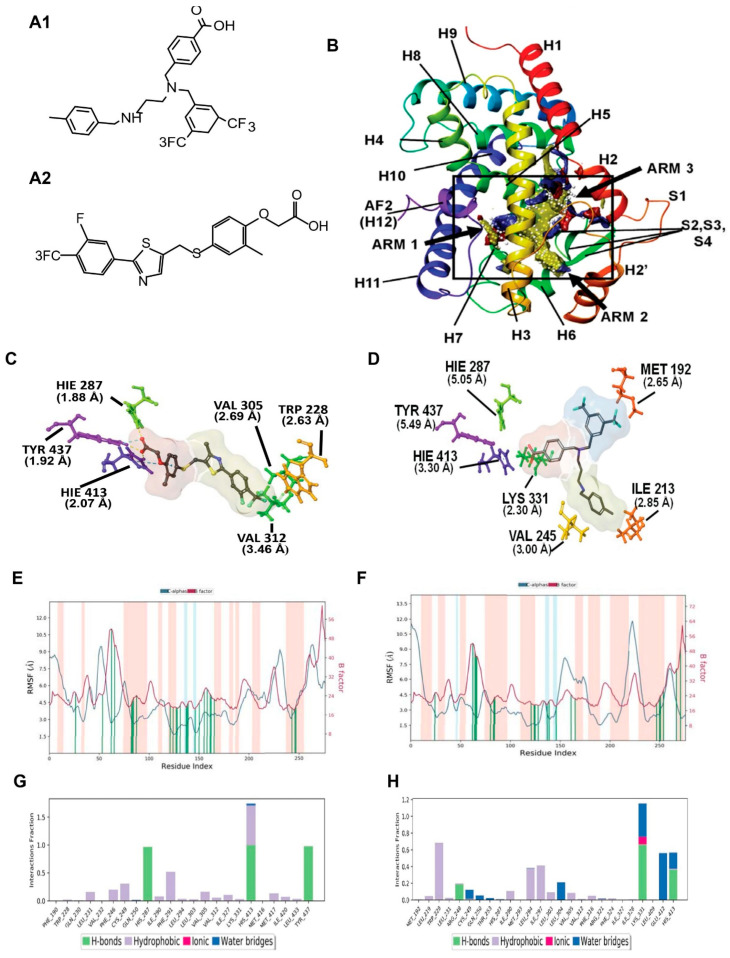
Molecular modeling for PPARδ showing comparison between GW0742 (PPARδ agonist) and AU9. (**A**) Chemical structure of AU9 (**A1**) and GW0742 (**A2**). (**B**) Site map analysis of PPARδ ligand binding domain (LBD) PDB: 3TKM. Hydrophobic surface map (yellow), hydrogen-bond donor surface map (blue), and hydrogen-bond acceptor surface map (red). α-helices and β-sheets labeled H1–H12 and S1–S4, respectively, from N-terminus to C-terminus. Y-shaped ligand binding pocket where arm 1 contains the AF2 domain, arm 2 is the entrance site, and arm 3 is a secondary ligand binding pocket: (**C**) GW0742 lowest energy conformation and amino acid binding interactions with distances (angstroms). GW0742 forms a hydrogen bond network to the AF2 domain, indicating a full agonist. (**D**) AU9 lowest energy conformation and amino acid binding interactions with distances. AU9 avoids a key interaction at TYR 437 (H12/AF2) yet maintains a critical contact at HIS 413. (**E**,**F**) Molecular dynamics root-mean-square fluctuation (RMSF) plots. The blue graph represents the ligand-induced α-carbon fluctuation overlaid with the red graph experimental b-factor. α-helices and β-sheets are shaded red and blue, respectively, from N-terminus to C-terminus. Vertical green bars indicate ligand-residue contacts. (**E**) GW0742 displays ligand-induced stabilization of the AF2 domain (residue index > 250) represented by a decrease in the RMSF as compared to the b-factor plot, approximately 5-fold. (**F**) AU9 displays ligand-induced stabilization of the AF2 domain while avoiding contact to TYR 437, approximately 1.5-fold. (**G**,**H**) Protein interaction diagram categorized by the fraction and type of interactions maintained throughout the simulation. (**G**) GW0742 maintains hydrogen bonds to the AF2 residue TYR 437 with supporting hydrogen bonds to HIS 413 and HIS 287 as the major contribution of ligand–protein contacts. (**H**) AU9 maintains a mixture of hydrogen bonds/water bridges/ionic interactions at LYS 331, GLU 412, and HIS 413, which predominate through the course of simulation. Analysis of the molecular dynamic simulation are reported in a protein–ligand contacts plot (**E**–**H**), which calculates the nature and fraction of bonds formed with protein residues throughout the simulation. Stacked bar charts are normalized over the course of trajectory, and values greater than 1 indicate that the protein residue is making multiple ligand contacts of different subtypes. The contributions a ligand may have on protein stability were calculated using the protein root mean square fluctuation (RMSF) plot. The protein RMSF plot ((**E**,**F**), Figure 2E,F) characterizes the local changes along the protein chain, relative to the ligand, throughout the course of simulation and is listed on the left-hand y-axis. The y-axis is the experimental x-ray B-factor. B-factors are experimentally determined from data submitted with the PDB x-ray crystal structure and indicate the relative vibrational motion with different atoms located in the structure [34]. Ligand induced changes in the protein RMSF should approximate the experimental B-factor, otherwise significant structural changes are occurring. Green vertical lines represent ligand–protein contacts. Shaded areas represent protein secondary structures, where red are alpha helices and blue are beta-strands ((**G**,**H**) and Figure 2G,H).

**Figure 2 cells-12-01116-f002:**
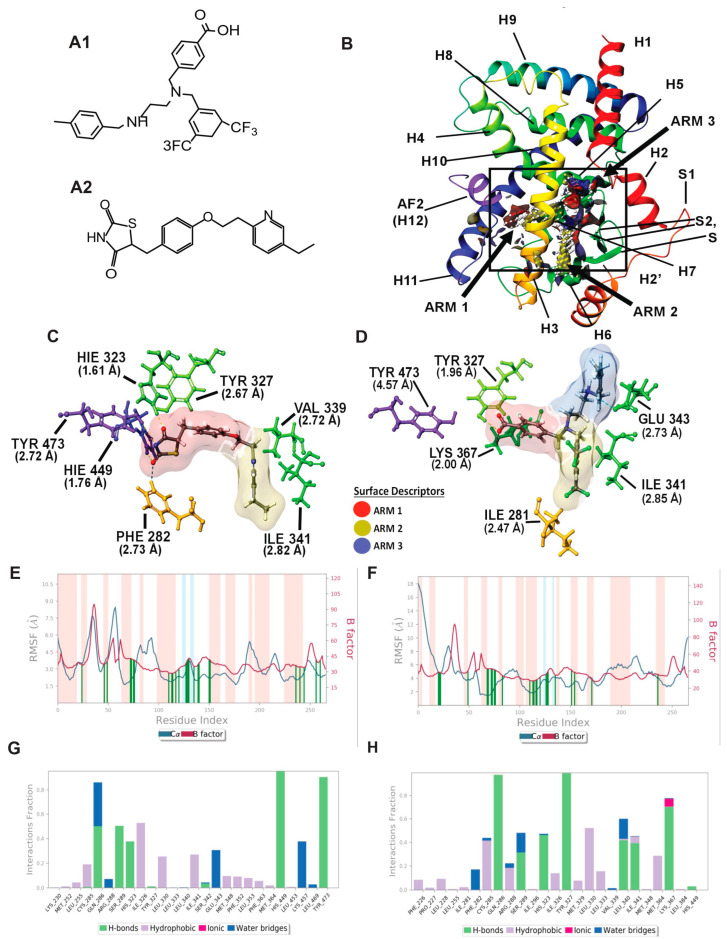
Molecular modeling: In silico modeling of PPARγ interactions between Pio (PPARγ agonist) and AU9. (**A**) Chemical structure of AU9 (**A1**) and Pio (**A2**). (**B**) Site Map analysis of PPARγ LBD PDB: 5Y2O. Hydrophobic surface map (yellow), hydrogen-bond donor surface map (blue), and hydrogen-bond acceptor surface map (red). α-helices and β-sheets labeled H1–H12 and S1–S4, respectively, from N-terminus to C-terminus. Y-shaped ligand binding pocket where arm 1 contains the AF2 domain, arm 2 is the entrance site, and arm 3 is a secondary ligand binding pocket. (**C**) Lowest energy conformation for Pio and amino acid binding interactions with distances (angstroms). Pio forms a hydrogen bond network to the AF2 domain indicative of a full agonist. (**D**) AU9 lowest energy conformation and amino acid binding interactions with distances. AU9 avoids a key interaction at TYR473 as its branched molecular structure inhibits extension further into the AF2 as compared to Pio. (**E**,**F**) Molecular dynamics root-mean-square fluctuation (RMSF) plots. Where the blue graph represents the ligand induced α-carbon fluctuation overlayed with red graph experimental b-factor, α-helices and β-sheets are shaded red and blue, respectively, from N-terminus to C-terminus. Vertical green bars indicate ligand–residue contacts. (**E**) Pio displays ligand-induced stabilization of the AF2 domain (residue index > 250) represented by a decrease in the RMSF as compared to the b-factor plot. (**F**) AU9 displays ligand induced stabilization in the global protein structure characterized by a reduction in RMSF as compared to the b-factor; however, avoidance of AF2 interactions cause a greater fluctuation in both the N-terminus and C-terminus indicating partial activity relative to a full agonist. (**G**,**H**) Protein interaction diagram categorized by the fraction and type of interactions maintained through the course of simulation. (**G**) Pio maintains hydrogen bonds to the AF2 residue TYR 473 with supporting interactions at residues HIS 449, HIS 323, SER 289, and GLN 286 as the major contribution of ligand-protein contacts. (**H**) AU9 maintains only minimal contact to the AF2 supporting residue HIS 449, thus contributing to the increased RMSF values observed in this region of the protein.

For comparison, the full PPARδ agonist GW0742 was used to illustrate key differences in our compounds’ ability to achieve a similar transcriptional activity in vitro. The protein–ligand contacts plot for GW0742 shows interactions occurring with twenty-three amino acid residues in the active site. The largest fraction of interactions occurring >50% throughout the simulation are hydrogen bond contacts to tyrosine 437 (H12), histidine 413 (H11) and histidine 287 (H5). This hydrogen bond network to the AF2 is characteristic of full PPAR agonists across all three isoforms, as ligand interactions with the AF2 lock the protein complex into an active conformation towards the recruitment of co-activators for gene transcription. The remaining contacts are primarily hydrophobic and occur outside of the AF2 domain, as listed in Figure 1G. The protein RMSF for GW0742 shows increased protein stability with all ligand contacts compared to the B-factor, except ligand contacts in H7. The protein–ligand contacts plot for compound AU9 shows interactions occurring with twenty-six amino acids in the active site. The largest fraction of interactions occurring >50% throughout the simulation is a mixture of hydrogen bonds, water bridges, ionic bonds, and hydrophobic bonds to lysine 331, tryptophan 228, glutamate 412, and histidine 413. Protein RMSF for AU9 shows increased stability with all ligand contacts compared to the B-factor, except contact between H1 and H2 (Figure 2B).

These results highlight the importance of PPARδ’s requirement for hydrophobic contacts from the ligand to provide stabilization of the protein complex. Compared to GW0742, AU9 forms fewer hydrogen bond interactions in the PPARδ LBD AF2. However, hydrogen bonding interactions to histidine 413 and a water bridge to glutamine 412 demonstrate a comparable stabilization of the AF2 domain. Additionally, the AU9 ligand interactions extend into arm three and provide stability in other regions of the protein that differ from GW0742.

PPARγ Induced Fit Docking and Molecular dynamics: The PPARγ LBD active site is similar to PPAR δ, as it also consists of a Y-shaped binding pocket. However, PPARγ has a decreased hydrophobic surface area and an increased polar surface area compared to PPARδ (Figure 3, PDB: 5Y2O). Entrance to the AF2 LBD of PPARγ can accommodate bulkier polar functional groups commonly seen in the thiazolidinedione (TZD) class of PPARγ selective agonists, thus providing substrate specificity [35]. Although full PPARγ agonists profoundly improve blood glucose levels, TZDs are associated with increased edema and heart failure [36]. Thus, AU9 was designed to avoid specific interactions in the PPARγ AF2 domain with the intention of improving clinical efficacy. Pio was used as the cognate ligand in this study to evaluate AU9′s partial agonist profile to that of a full agonist. Pio forms a strong hydrogen bond network to the AF2 domain via contacts at tyrosine 473 (H12), histidine 323 (H4), and histidine 449 (H3) (Figure 3A). Supporting hydrogen bonds from phenylalanine 282 and tyrosine 327 help to provide further stability for the AF2 domain. Pio’s lipophilic tail extends into arm 2 to make hydrophobic contacts at valine 339 and isoleucine 341. In comparison, AU9 avoids contact with tyrosine 473 as its branched structure prevents extension into arm 1. AU9 forms a hydrogen bond to tyrosine 327 and lysine 367, providing partial stabilization towards the AF2 domain. Further hydrophobic contacts to glutamate 343, isoleucine 341, and isoleucine 281 provide comparable stabilization of the protein complex to that of Pio (Figure 3B).

The protein–ligand contacts plot for Pio shows interactions occurring with twenty-six amino acid residues in the active site. The largest fraction of interactions occurring >50% throughout the simulation are hydrogen bonds to tyrosine 473 (H12/AF2), histidine 449 (H11), serine 289, glutamine 289, and a hydrophobic contact at isoleucine 326 seen in Figure 2G.

The protein–ligand contacts plot for AU9 shows interactions with twenty-five amino acids within the PPARγ LBD. The interactions occurring >50% of the simulation are hydrogen bonds to glutamine 286, tyrosine 327, lysine 367, and leucine 340 (Figure 2H).

All ligand contacts observed in the protein RMSF plots for both Pio and AU9 (Figure 2E,F) show increased stability to that of the B-factor plot, except the AF2 domain. Pio ligand contacts to the AF2 were shown to decrease the protein RMSF relative to the B-Factor. Conversely, AU9 avoids contact with the AF2 LBD tyrosine 473 residue, thereby allowing for greater AF2 flexibility and dynamic motion. As a consequence, it would be expected that AU9 would display a partial agonistic profile to that of Pio in vitro.

PPARδ and γ reporter assay: The ability of AU9 to induce PPARδ and PPARγ activity was determined by zLBD-Driven GAL4 Reporter assay (Figure 3A,D). Values obtained from the dose–response curve (Figure 3A) suggest that AU9 has PPARδ activity when compared to GW0742 (EC_50_ of AU9 is 41 nM and EC50 for GW0742 is 20 nM). Further evaluation of AU9 activating PPARδ is observed by the promoter activity (Reporter) assays involving AU9 inducing interaction of PPARδ with the PPARδ response element (DRE) similar to GW0742 (10 fold from control), a full PPARδ agonist (Figure 3B) (*p* < 0.0001). Additionally, our animal studies confirmed that AU9 induced an increase in PPARδ gene expression targets, as demonstrated by the PPARδ downstream gene expression profile observed by a bubble diagram (Figure 3C). Conversely, AU9 did not have a significant effect on PPARγ activation (Figure 3D) or the constitutively active (VP16) PPAR gamma construct (Figure 3E,F). EC50 is respectfully measured at 400 nM (AU9) and 50 nM (pioglitazone) (*p* < 0.05 and *p* < 0.0001). Further evaluation of AU9 effects on the AP2 promoter further confirmed that AU9 activates PPARγ (Figure 3E). To help explain this, our in silico design predicted that AU9 avoids Tyrosine-473 of the AF2 ligand binding domain of PPARγ. Therefore, we tested and observed that the substitution of Tyrosine-473 with phenylalanine resulted in a significant reduction of Pio-mediated activation of the PPARγ interaction with PPRE (1.9 fold) (Figure 3F,G) (*p* < 0.05 and *p* < 0.0001).

AU9 improves deficits in Y-maze and NOR tests in 3xTg-AD mice: The activation of the PPARδ and PPARγ axes improves cognitive deficits in mouse models for AD [37,38]. We hypothesized that AU9 may play a role in synaptic processes and ultimately cognition and that selective activation of PPAR by AU9 may improve learning and memory deficits. To determine whether AU9 improves cognitive deficits in 3xTgAD mice, we performed novel object recognition (NOR) and Y-maze based on previously published protocols by our group and others, as discussed in the methods section [16]. In the NOR test, there was no biased exploratory preference to either object among the four groups of mice in the training session, suggesting that there was no difference in motivation and curiosity about novel objects among the groups (data not shown). In the retention session performed 24 h after training, we observed a marked decrease in the exploratory preference (Figure 4A) for novel objects, as observed in saline-treated 3xTgAD mice compared with that in control mice (reduced by ~20%, *p* < 0.05 and *p* < 0.001), indicating impaired discrimination of a novel object from a familiar one (Figure 4B). Furthermore, treatment with AU9 significantly improved the discrimination index in comparison to the vehicle-treated 3xTgAD group (*p* < 0.001). These results suggest that AU9 improves recognition memory impairment in 3xTgAD mice by two folds (*p* < 0.001).

To determine the effect of AU9 on short-term spatial recognition memory, we utilized a two-trial Y-maze task with an inter-trial interval of 3 h. The number of arm entries and the time spent in the novel arm was significantly less by the 3xTgAD mice when compared with the control mice (Figure 4D–F) (*p* < 0.001) and that AU9-treated 3xTgAD mice demonstrated an improvement in the number of entries in the novel arm (Figure 4F, *p* < 0.05). These results suggest that AU9 improves short-term memory impairment in 3xTgAD mice.

Field recordings in 3XTg-AD mice: To determine whether cognitive impairment in the 3xTgAD mice was linked to alterations in neural field transmission, hippocampal slices were used to measure the fEPSP responses at increasing stimulus intensities based on previous protocols [19,39]. We observed an alteration in fEPSPs over a range of stimuli intensities between groups. The fEPSP slope and amplitude were reduced in 3xTgAD mice compared to the controls (Figure 5A,B) (*p* < 0.001). Further, there was an improvement in transmission following AU9 treatment (Figure 5A) (*p* < 0.05). To further investigate whether the deficits in transmission in 3xTgAD mice were potentially due to alterations in presynaptic axon recruitment, we measured the fiber volley (FV) amplitude across a range of increasing stimulus intensities (amplitude) (Figure 5B). We observed that 3xTgAD mice showed a reduction in the FV amplitude compared to the wild type mice, suggesting reduced presynaptic axonal activation/recruitment. However, no improvement in FV amplitude deficit was observed in AU9 treated 3xTgAD mice (Figure 5B).

Field recordings in 3xTgAD mice: We next examined whether cognitive impairments in 3xTgAD were associated with alterations in neuronal activity by measuring LTP, based on previous methodologies [16,19,40]. Using hippocampal slices, we determined that 3xTgAD mice displayed deficits in LTP in the Schaeffer collateral pathway when compared to the control mice. The 3xTgAD mice treated with AU9 showed an improvement in the fEPSP and LTP (Figure 5C,D) (*p* < 0.0048). One possibility for the reduction LTP can be attributed to weakened signaling strength during LTP [41]. To evaluate for alterations during LTP induction (Figure 5E) (*p* < 0.001), we assessed fEPSP amplitude during theta burst stimulation and observed a significant difference between the control and 3xTgAD. Further, there was no significant difference between the control and 3xTgAD mice treated with AU9. These results suggest that AU9 improves the deficits in LTP in the hippocampus of 3xTgAD mice.

AU9 improves neurotrophin levels and spine density: Hippocampal function in the form of neuronal survival and differentiation is primarily dependent on neurotrophins, including brain-derived neurotrophic factor (BDNF) [42,43]. Neurotrophins are required for supporting the synapse-specific protein synthesis that mediates the stability of various forms of synaptic plasticity [43,44]. Several studies have indicated reduced BDNF and neurotrophin levels in the brains of patients diagnosed with AD and mild cognitive impairment (MCI) [45,46,47]. Similarly, reduced BDNF levels are also observed in animal models of AD [48]. Previous findings from our lab have established that the PPARγ agonist rosiglitazone promotes BDNF gene expression [40]. Hence, we sought to investigate whether AU9 could improve neurotropin levels, including BDNF expression, in 3xTgAD mice. To validate our theory., we measured neutropin levels via an ELISA and observed a significant increase in neurotropin levels following AU9 treatment in 3xTgAD mice in comparison to saline treated 3xTgAD mice (Figure 6A–D) (*p* < 0.05). Interestingly, we noted a statistically significant increase in neurotrophins in 3xTgAD treated with AU9 for 3 months including BDNF (increase by ~10 pg/mg of protein) (*p* < 0.05), Glial-derived Neurotrophic Factor (NGF) (increase by ~10 pg/mg of protein) (*p* < 0.005), and NT3 (increase by ~10 pg/mg of protein) (*p* = 0.966, and nt 4/5 (*p* > 0.05) levels (Figure 6A–D). Taken together, our data suggest that AU9 treatment improves neurotrophin levels in 3xTgAD mice. Neurotrophins can promote dendritic spine morphogenesis, including improved spine density, area, and length. We observed statistically non-significant improvement in spine density as determined from our Golgi-Cox staining results (Figure 6E–H).

### 3.2. AU9 Reduces Aβ Levels in 3xTgAD Mice

The 3xTg-AD mice develop amyloid plaques by six months of age. The pathologies appear in a distinct pattern, with Aβ deposition starting in the neocortex and appearing later in the hippocampus [49]. Immunostaining for Aβ with 6E10 antibody revealed significantly increased Aβ deposits in the hippocampi of vehicle-treated 3xTgAD mice compared to AU9-treated mice (Figure 7A,B). Specifically, the detectable Aβ levels were markedly reduced in the hippocampi of 3xTgAD mice treated with AU9 compared with vehicle-treated mice (Figure 7A) (0.85 fold) (*p* < 0.004). Further analysis of the soluble form was measured by an Aβ1–42 enzyme-linked immunosorbent assay (ELISA). We observed a statistically significant reduction in Aβ1–42 levels in the 3xTgAD mice treated with AU9 when compared to saline-treated mice (decrease by eight pg/mg total protein). We confirmed our findings in APP-Cho cells and observed that AU9 has to reduce Aβ levels by approximately 50% (10 µM); (Figure 7C) (*p* < 0.001). Several studies have indicated that PPAR agonists reduce BACE1 expression and thereby reduce Aβ levels [24]. We, therefore, investigated the effect of AU9 on BACE1 activity and found that AU9 reduces Aβ in our APP-Cho cell line and reduces β-secretase activity with increasing concentrations of AU9 (Figure 7D) (*p* < 0.001 and *p* < 0.0001).

### 3.3. AU9 Reduces Neuroinflammation

Previous reports verify an increase in neuroinflammation associated with an increase in marker cells (microglia) as well as infiltrating macrophages. We investigated changes in gene expression patterns using gene analysis (qPCR) and Nanostring data analysis (Appendix A). Changes in gene expression patterns associated with neuroinflammation and cytokine expression verify that AU9 attenuated several markers associated with neuroinflammation in 12-month-old 3xTgAD mice (Figure 8A). Further, the markers IBA 1 and TSPO were observed to increase in 3xTgAD mice brains, approximately 0.7- and 1.75-fold increases from wild type mice (*p* < 0.05) and (*p* < 0.05), respectively. Further, AU9 (5 mg/Kg for 3 months daily) significantly reduced IBA expression (0.5-fold and 1.70-fold, respectively, *p* < 0.05) in 12-month aged 3xTgAD mice. Further nano-string analysis allowed us to determine that AU9 treatment in the same mice resulted in reduced cytokine expression cellular stress and DNA damage (Appendix A). Lastly, we measured in BV2 cells that AU9 reduced lipopolysaccharide mediated nitrite levels in a dose response manner (0–100 μM) where a 50% reduction was observed at a dose of 10 μM of AU9 (*p* < 0.0001).

### 3.4. Peripheral Effects of AU9

Traditional full PPARγ agonists are known to induce an increasing body weight. However, PPARδ agonists are known to improve oxidative phosphorylation and catabolic activity. After 3 months of treatment in 9- to 12-month-aged mice, we observed no significant increase in body weight in both wild type and 3xTgAD mice (Figure 9A,B). Further, it has been reported that 3xTgAD mice display elevated blood glucose levels when compared to age-matched wild type mice [50]. We observed a significant improvement in our glucose tolerance test in 12-month-old 3xTgAD mice (Figure 9C) (*p* < 0.001 and 0.05). Lastly, AU9 (10 mg/kg) did not induce a significant increase in heart weight to body weight ratio (0.25 fold increase in wild type and 0.27 in 3xTgAD mice). However, we observed a significant increase with Pio (10 mg/Kg) in wild type and 3xTgAD mice after 3 months of treatment (Figure 9D) (*p* < 0.05 and *p* < 0.001).

## 4. Discussion

PPARγ agonists have previously been investigated as potential treatments for Alzheimer’s disease; however, there have been conflicting data from preclinical studies. For example, full PPARγ agonists Pio and rosiglitazone (rosi) improved cognition in the PS1-KI (human presenilin-1 M146V knock-in mouse) mouse model of AD following a 9-month treatment (20 mg/kg) [10]. However, in 3xTgAD mice, similar effects were not observed; thus bringing into question the ability of Pio to improve memory deficits in AD [10,51]. Additionally, object recognition studies revealed a trend towards the worsening of memory in wild type male mice after Pio treatment, thus making the overall effect of Pio on cognition difficult to interpret [10]. However, evidence suggests that targeting PPARγ and/or PPARδ can improve memory deficits and/or the pathology associated with Alzheimer’s disease in rodent models. Indeed, work by Searcy et al. demonstrated that pioglitazone in 10-month-old 3xTgAD mice improved learning on the active avoidance task, decreased hippocampal amyloid-β and tau deposits, and enhanced short- and long-term plasticity [52]. Interestingly, in human diabetic patients with mild cognitive deficits, Pio treatment improved peripheral insulin sensitivity, as well as plasma levels of Aβ and the insulin degrading enzyme [53]. However, findings from the TOMMORROW clinical trial utilizing pioglitazone in patients with MCI failed to improve cognitive deficits [54]. These confounding results suggested that an alternative form of PPAR agonism would have significant potential for AD. However, work by Joel Berger’s group at Merck Pharmaceutical identified the significance of the physical interactions of Tyrosine-473 in the PPARγ ligand binding domain for adiposity and other biological properties [55]. Secondly, our previous work identified PPARδ as a potential therapeutic target for improving synaptic plasticity in rodent models of diabetes/AD [38]. We therefore developed a dual PPARδ-PPARγ agonist. Our in silico observations of full PPAR agonists have been shown to form strong hydrogen bond interactions with the AF2 ligand binding domain contained in arm 1. Further stabilization of the ligand–protein complex is achieved by hydrophobic interactions in arm 2. The ligand’s ability to form a stable hydrogen bond network to the AF2 is representative of a strong transcriptional activation, as this leads to the displacement of co-repressor proteins and recruitment of co-activator proteins. As the AF2 adopts this active conformation, changes in the quaternary protein structure allow for new sites to become available for co-activator binding. Ligand-induced conformational changes can affect the size and residue charge distribution to accept a variety of co-activators. However, due to the functional difference in the two arms seen in PPARs, different binding modes can be adopted with some ambiguity to the specific role the arms play during transcriptional activation. Interestingly, this presents the potential for novel mechanisms of activation in PPARδ ligand binding. While AU9 does not display the typical binding profile of a full agonist, it does have the ability to provide significant stabilization of AF2 domain at histidine 413 on H11 (Figure 1B). Coordination of H11 alone can position the H12 AF2 domain for self-assembly of tyrosine 437 to histidine 287 in the absence of ligand stabilization, providing some explanation for the observed activity of AU9. Furthermore, AU9′s branched structure extends deep into arms 2 and 3, providing several stabilizing hydrophobic contacts at valine 245, isoleucine 213, and methionine 192 (Figure 1B). The combined space filling of arm 2 and 3 by AU9 provides additional stabilization of the protein complex outside of the AF2, which is not observed in the PPARδ agonist GW0742.

It is interesting to note that the trifluoro side group interaction holds the ring moiety in position, making AU9 avoid contact with Tyr473, which is approximately 5Å away. This residue is crucial to the stabilization of the AF2 helix H12, which allows the binding of co-activators that lead to the activation of the genes responsible for adipogenesis [55]. Our in silico data were confirmed by transcriptional assays, which demonstrate that AU9 minimally activates the PPARγ AP2-PPRE (Figure 4A). The results appear consistent with our previously published lipid accumulation assays, where AU9 has negligible effects upon lipid accumulation in adipocytes [14].

Ligand binding affects the conformation of the AF-2 ligand binding surface, resulting in modifying the binding affinity for chromatin remodeling transcriptional co-regulator proteins and resulting in the activation or repression of selective gene transcription [56,57]. Further evidence for conformational changes associated with ligand–receptor interactions has been identified by crystal structures that define the inactive/repressive and active conformations that enable the binding of transcriptional coactivator and corepressor proteins, respectively, by stabilizing specific conformations of the AF-2 region [58]. Recent work has illuminated how ligands engage the ligand-binding domain and enter the orthosteric ligand-binding pocket, and whether ligand binding occurs through an induced fit or conformational selection mechanisms [59]. In the induced fit scenario, ligand binding selectively binds to and selects a particular conformation that is occupied within the ligand binding conformational group. In the conformational selection mechanism, ligand binding occurs through an encounter complex and promotes the ligand binding conformational group into the final ligand–bound complex.

Previously, we have observed that PPARδ agonist (GW0742) and rosiglitazone improve synaptic plasticity in db/db leptin receptor knockout mice [38,40]. The key findings of the current study are that 3xTgAD mice display cognitive deficits and impaired synaptic plasticity that can be rescued by AU9 through activation of PPARδ and partially PPARγ. This conclusion is deduced mainly from the following observations. First, AU9 treatment improved cognitive deficits, specifically impairments in working memory (assessed by NOR and Y-maze) in 3xTgAD mice. Second, hippocampal LTP was impaired in 3xTgAD mice, and AU9 improved the deficits. Third, AU9 modulated the postsynaptic receptor expression in the hippocampus of 3xTgAD mice. Importantly, AU9 improved neurotrophin levels including BDNF levels in the hippocampus. Fourth, AU9 modulated several hippocampal genes involved in synaptic plasticity and neurotransmission.

After 12 weeks of treatment, novel object recognition performance and Y-maze-dependent memory were improved in AU9-treated 3xTgAD mice. This is in agreement with prior studies, which reported that the PPARγ agonist rosiglitazone attenuated the learning and memory deficits of APP transgenic mice in the radial maze and object recognition tests after chronic administration [5,6]. However, APP mice treated with Pio for two months did not have improved spatial memory [53]. The difference in the latter study may be due to the different dosing of the agonists used and the duration of the treatment.

Several animal models of AD exhibit deficits in basal synaptic transmission, which correlate with the progression of the disease [60]. Some of these deficits include, the release of glutamate, glutamate uptake, and the expression or functionality of glutamatergic receptors [61]. Furthermore, neurotrophins, including BDNF have been shown to regulate the expression and synaptic delivery of AMPA receptor subunits in the hippocampus, which implies that neurotrophin signaling alters AMPAR trafficking in addition to influencing AMPAR activity [62]. Therefore, the improved neurotrophin expression may be responsible for the improved basal synaptic transmission in part via influencing AMPAR trafficking and function [39]. Further evidence for improved markers associated with neurotransmission can be seen in our gene analysis profile (nanostring data) in Appendix A.

In the present study, we observed deficits in LTP, an integral component of the signaling strength and synaptic dysfunction observed in AD pathology [63]. We also found that AU9 enhances LTP in the hippocampal Schaffer collateral pathway in the 3xTg-AD mice without affecting the control LTP. Other studies have also reported an influence of PPARδ and PPARγ receptor signaling on LTP [7,38]. Previously, we observed an improvement in LTP deficits in mouse model of diabetes (leptin receptor deficient db/db) following administration of rosiglitazone via ICV and not by oral delivery [40]. PPAR agonists increase the expression of neurotrophins and transcription factors, including neurotrophic factor 1α or CREB [64], which are centrally involved in this process [40,65,66]. Neurotrophins are required for supporting synapse-specific protein synthesis that mediates the stability of long-term forms of synaptic plasticity [67]. Likewise, we observed an increase in four neurotrophins, including BDNF.

In the current study, the treatment of 3xTgAD mice with AU9 reduced hippocampal Aβ deposition compared to the control mice. We noted a significant decrease in the total Aβ plaque area and the respective staining intensity. Furthermore, soluble levels of Aβ were significantly reduced following AU9 treatment in 3xTgAD mice. Thus, AU9 may be involved in preventing the formation of Aβ deposits or augmenting the clearance of Aβ. Several lines of evidence indicate that PPARγ transcriptionally regulates the activity of beta secretase enzyme (BACE1), a key enzyme responsible for generation of Aβ peptides [68,69]. Hence, activating PPARγ with natural or synthetic ligands inhibits BACE1. We observed that AU9 inhibited BACE1 activity, which may explain the reduced Aβ levels in culture and our animal models.

Oxidative stress and damage are implicated in Alzheimer’s disease and are linked to Aβ plaque formation, Alzheimer’s disease pathophysiological events, and synaptic dysfunction [70]. Increased ROS occurs due to an imbalance between pro-oxidants (ROS, RNS, superoxide anion, hydroxyl radicals, and hydrogen peroxide) and antioxidants (GSH, GPX, CAT, GRx, and SOD). Down-regulation of antioxidant defense mechanisms and elevated ROS generation lead to oxidative stress-mediated neurodegeneration [71]. We observed a statistically significant increase in ROS levels in 3xTgAD mice, which were attenuated by the AU9 treatment. AU9 treatment in 3xTgAD reduced markers associated with neuroinflammation, markers of stress, and DNA damage, as seen in our Appendix A.

Our study had several shortcomings, including that our model was a moderately aged (9–12 months treatment age) 3xTgAD model. Further work on understanding long-term treatment into the late stage (16–18 months) will help understand whether AU9 can reduce or prevent the progression of the disease into late stages. Another limitation is the use of only female mice due to previous studies determining that the gene expression in males is unstable. Alternative Aβ models, such as the 5xFAD and a Tau model (P301S), may provide more insight into AU9′s ability to alter more aggressive forms of pathology. As postmortem brains from late stage AD patients indicate hyperinsulinemia and altered insulin signaling, determination of the impact of AU9 on brain glucose uptake and insulin signaling will help us better understand the impact of PPAR signaling on brain energy regulation. Our neurotrophin results offered a possible explanation for how improvement in our LTP—field recordings. Although we observed no significant influence of AU9 on presynaptic activity, further information determining the influence of AU9 on postsynaptic receptor involvement is needed. Potential studies would include patch clamp analysis and synaptosome fractional analysis for the influence of AU9 on glutamatergic and or NMDAR receptor levels in the postsynaptic region.

## 5. Conclusions

We have developed a novel PPARδ–partial PPARγ agonist that improves behavioral deficits and synaptic plasticity (LTP) in an AD mouse model. The anti-inflammatory effects and enhanced neurotrophin expression levels may help explain these findings. Further analysis to clarify how AU9 ameliorates amyloid beta levels needs to be further explored. Future pharmacodynamic and pharmacokinetic analyses will be beneficial to advance the clinical application of AU9. The current study’s findings support the potential use of PPAR agonists in the treatment of AD.

## 6. Patents

Patents: 10844003 Dual PPAR-delta and PPAR-gamma agonists to Drs. Amin and Ward and are licensed to Oleolive llc.

## Figures and Tables

**Figure 3 cells-12-01116-f003:**
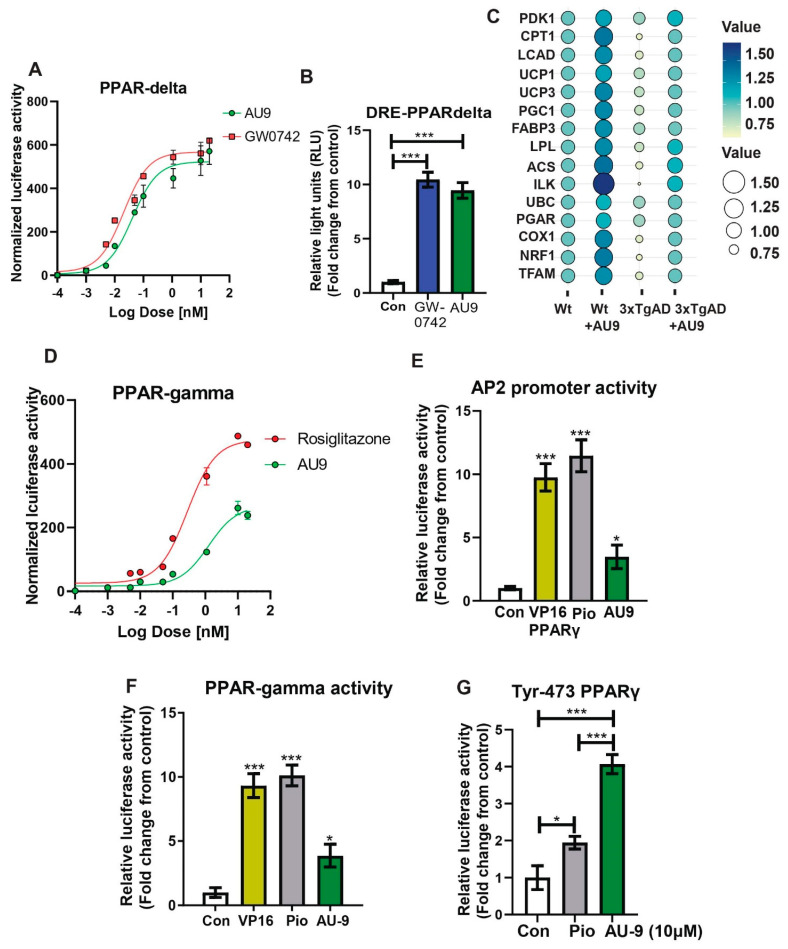
Reporter assays for AU9 PPARδ and PPARγ activity. Stable HEK293T cell lines expressing (**A**) PPAR-δ ligand binding domain driven GAL4 reporter assays determined AU9 activity when compared to increasing concentrations of full PPARδ agonist GW0742. (**B**) AU9 induces partial human PPAR-δ activity when compared to full PPARδ agonist GW074 by activating the PPARδ Response Element (DRE) via transient co-transfection into HEK293 cells with human PPARδ expression vectors along with the reporter plasmid (PPRE-pk-Luc) or control reporter plasmid (pk-Luc) with *Renilla* vector for 24 h. Cells were treated with AU9 (10 μM) and Pio (10 μM) for 24 h. Luciferase activity was normalized to *Renilla* luciferase activity as described in the Methods section. Values were based upon normalized luciferase activity and ΔΔct values shown as a fold change from control. Statistical values were obtained using two-tailed, unpaired *t*-test analysis ± S.E.M. Where *n* = 6 independent experiments with three replicates per experiment, *** *p* < 0.0001. (**C**) Bubble plot of qPCR analysis (fold change from control) of wild type and 3xTgAD mice treated with and without AU9 for three months daily (5 mg/Kg). (**D**) Stable HEK293T cell lines expressing PPARγ ligand binding domain driven GAL4 reporter assays determined AU9 activity compared to increasing concentrations of full PPARγ agonist Rosi. (**E**) AU9 induces human PPARγ activity by activating the AP2 response element via co-transfection into HEK293 cells transiently with human PPARγ vector with the reporter plasmid (AP2-Luc) or control reporter plasmid (pk-Luc) with *Renilla* vector for 24 h. (**F**) AU9 induces human PPARγ activity by activating the 3XPPRE-pk-Luc response element via co-transfection into HEK293 cells transiently control reporter plasmid (pk-Luc) with *Renilla* vector for 24 h. (**G**) Human PPARγ with tyrosine-473 substituted with phenylalanine demonstrates activity using the 3XPPRE-pk-Luc response element via co-transfection into HEK293 cells transiently control reporter plasmid (pk-Luc) with *Renilla* vector for 24 h. Cells were treated with AU9 (10 μM), and Pio (10 μM) for 24 h. Luciferase activity was normalized to *Renilla* luciferase activity as described in the Methods section. Values for G and H were based upon normalized luciferase activity and fold change from control. Statistical values were obtained using two-tailed, unpaired *t*-test analysis ± S.E.M. Where *n* = 6 independent experiments with three replicates per experiment. *: *p* < 0.05 and ***: *p* < 0.0001.

**Figure 4 cells-12-01116-f004:**
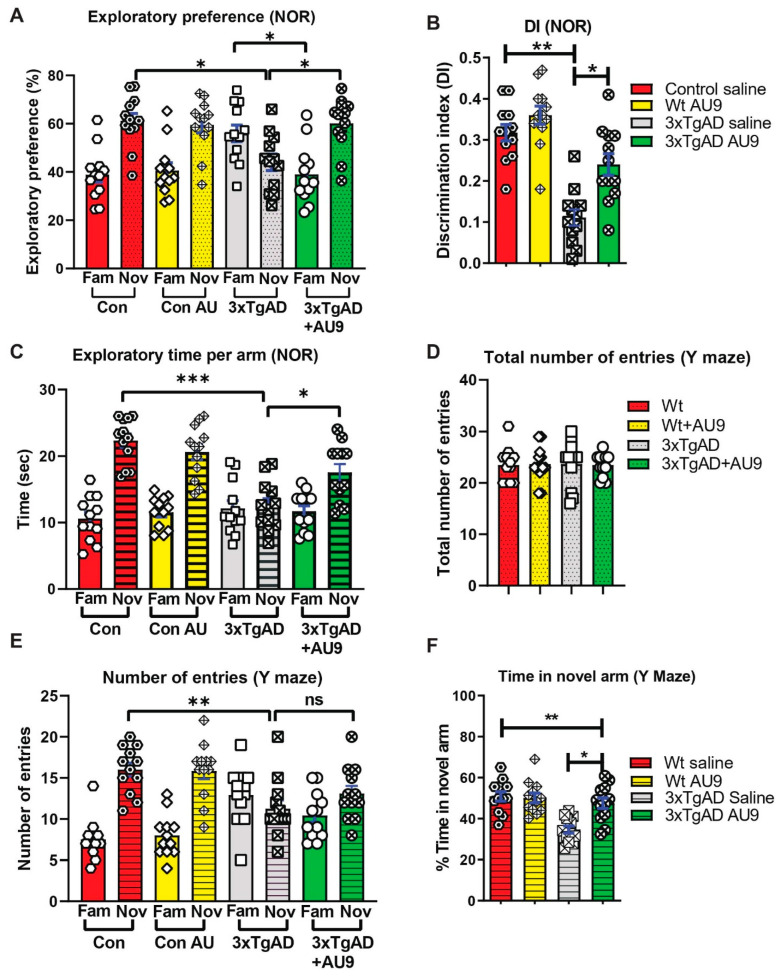
AU9 improves memory deficits in 3xTgAD mice. (**A**) Results for novel object recognition (NOR) tests reflect an exploratory preference for the novel object vs. the familiar object by the animals that were analyzed by a naive subject. (**B**) Discrimination index (DI), represents the recognition of memory sensitivity. Where discrimination index was calculated as (DI) (T novel − T familiar)/(T familiar + T novel). (**C**) The exploratory time of a novel object when compared to the familiar object. (**D**) Results from Y-maze tests for the total number of entries into the novel and familiar arms. (**E**) Results from Y-maze tests for the number of times in the novel or familiar (other) arm. (**F**) The Y-maze tests for the percent time the animal spends in the novel arm. Statistical values were obtained by student t test analysis ± S.E.M. Where *n* = 12 mice per group and ns represents no significance, where * *p* < 0.05, ** *p* < 0.001 and *** *p* < 0.0001.

**Figure 5 cells-12-01116-f005:**
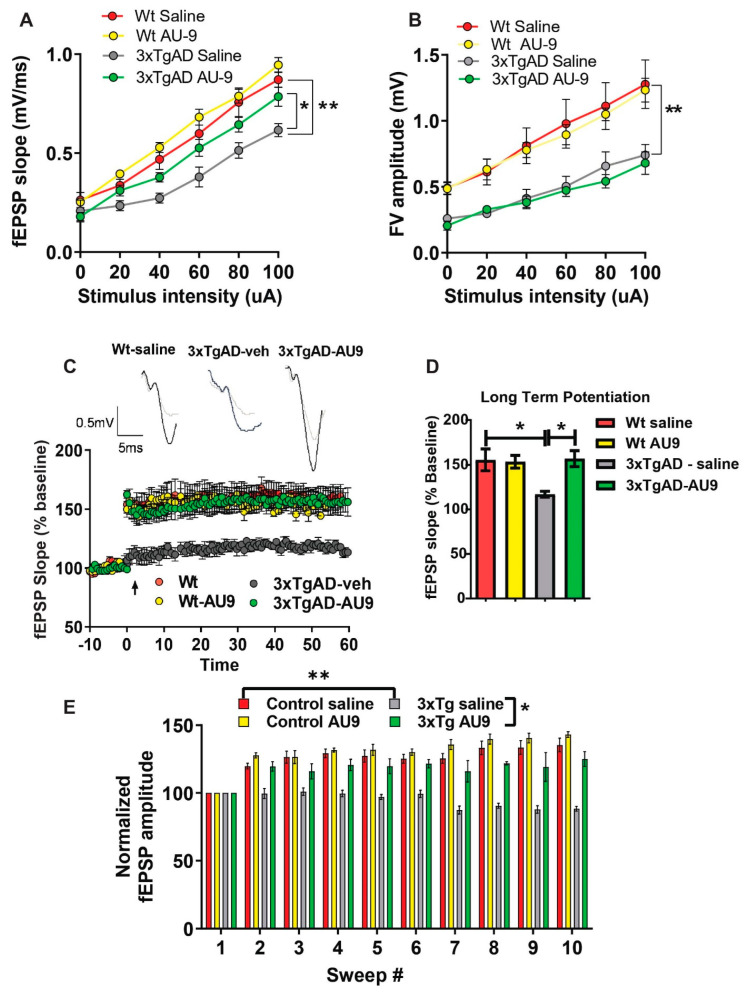
AU9 improves field recordings in 3xTgAD mice. 3xTgAD mice display alterations in extracellular field recordings of excitatory postsynaptic potential (EPSP). Twelve-month-aged female wild type and 3xTgAD mice treated with AU9 or saline orally for 3 months daily (5 mg/kg). (**A**) Input–output curve of fEPSP slope recorded at increasing stimulus intensities. (**B**) Input–output curve of FV amplitude recorded at increasing stimulus intensities. (**C**) Deficits in 3xTgAD mice LTP was improved in AU9-treated 3xTgAD mice as measured by a high-frequency stimulation (3 × 100 Hz trains with a 20 s intertrain interval). LTP graphs represent fEPSP slope before and after induction by TBS. (**D**) LTP bar graphs show fEPSPs recorded during the time period 50–60 min following TBS induction normalized to baseline levels and traces before and after LTP induction. (**E**) Sweep analysis was calculated by normalizing the amplitude of the first fEPSP of sweeps 2–5 with the amplitude of the first fEPSP of sweep 1 during LTP induction. Statistical values were obtained using student *t*-test analysis ± S.E.M. Where *n* = 8 mice per group and *: *p* < 0.05, **: *p* < 0.001.

**Figure 6 cells-12-01116-f006:**
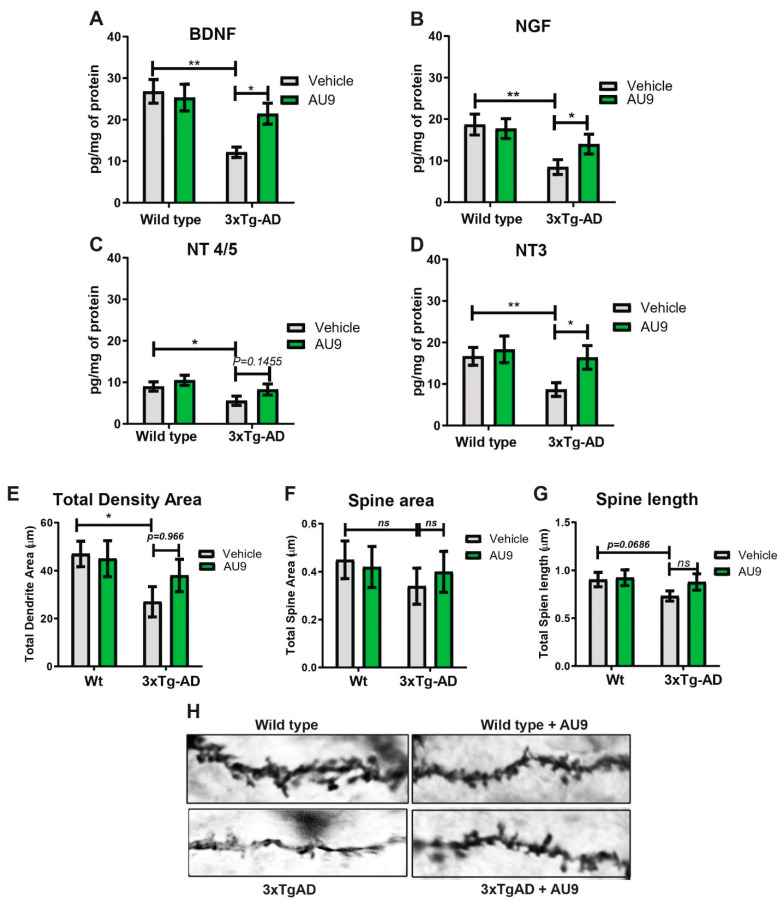
AU9 improves neurotrophin expression and spine density. AU9 improves neurotrophin protein expression in 3xTgAD mice treated with AU9 (3 months daily, 5 mg/kg) as determined by ELISA, (**A**) Brain-Derived Neurotrophic Factor (BDNF) (**B**) Nerve Growth Factor (NGF) expression, (**C**) Neurotrophin Factor 4/5 expression and (**D**) Neurotrophin −3 expression in 3xTgAD hippocampi. Where *n* = 6 mice per group of treatment and * *p* < 0.01, and ** *p* < 0.005. Values were based upon a normalized protein concentration, a standard curve of growth factor protein supplied in the kit. Statistical values were obtained using Student *t*-test analysis ± S.E.M. Where *n* = 6 mice per group and *: *p* < 0.05 and **: *p* < 0.001. AU9 improves spine density area, spine area and spine length (**E**–**H**) in 3xTgAD mice. Rapid Golgi-Cox staining was utilized to measure changes in total spine density, spine area and spine length. Overall, 200 µm sections were stained and imaged using a Z-stack procedure on a Nikon TSi microscope from a minimum of 10 slices with 10 neurons per slice from 6 mice per group using ImageJ software for measurements. Statistical values were obtained using student *t*-test analysis ± S.E.M. Where n = 6 mice per group and *: *p* < 0.05.

**Figure 7 cells-12-01116-f007:**
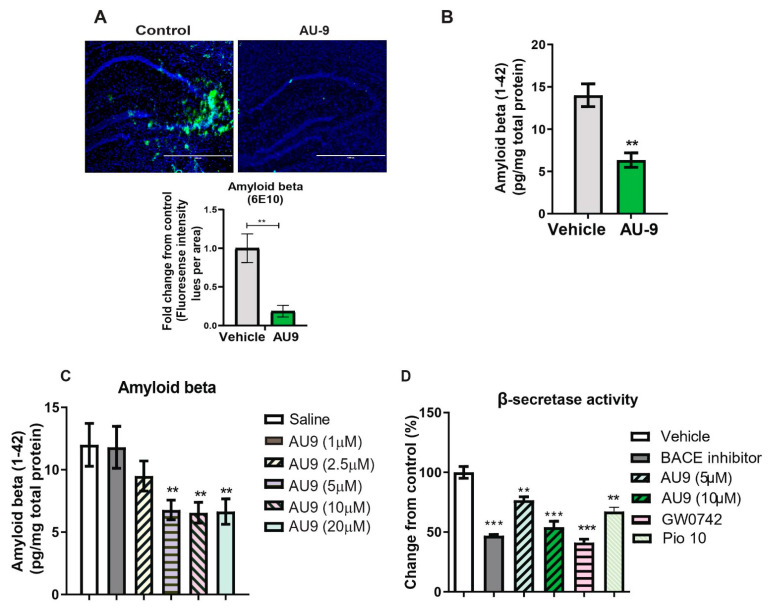
AU9 reduces amyloid beta (1-42) levels in 3xTgAD mice. (**A**) Immunofluorescence imaging (anti-6E10 antibody) shows a reduction in levels of all forms of Amyloid beta levels in 12-month-old 3xTgAD mice treated with AU9 orally for three months (5 mg/Kg daily). Densitometric measurements of Amyloid beta in hippocampi from six mice and four slices per mouse. Values were standardized to total area. (**B**) Elisa measurement of soluble form of Amyloid beta (1–42) from mice treated in the same manner as mice as in panel A. Amyloid beta was measured from hippocampi from six mice per group. Values were based on a standard curve of Amyloid beta 1–42 and standardized to total protein concentrations. Statistical values were obtained using Student *t*-test analysis ± S.E.M. Where *n* = 6 mice per group and ** *p* < 0.001. (**C**) Reduction of Amyloid-beta being secreted in media from APP-Cho cells following increasing concentrations of AU9 treatment (1, 2.5, 5, 10 and 20 µM). Values were based upon standardized curve from ELISA (R&D Systems). Statistical values were obtained using student *t*-test analysis ± S.E.M. Where *n* = 6 independent experiments were repeated in triplicate per group and **: *p* < 0.001. (**D**) Effects of AU9 (5 µM and 10 µM) on Beta secretase activity in APP-Cho cells. β-Secretase activity was determined fluorometrically using an β-Secretase activity kit (Biovision, Waltham, MA) and standardized to total protein concentration from APP-Cho cells. Beta-secretase activity was represented as relative fluorescence unit per mg of total protein. Values were based upon means from 6 independent repetitions with 3 replicates in each group and represented as a percent change from control. Statistical values were obtained using student *t*-test analysis ± S.E.M. **: *p*<0.001, ***: *p* < 0.0001.

**Figure 8 cells-12-01116-f008:**
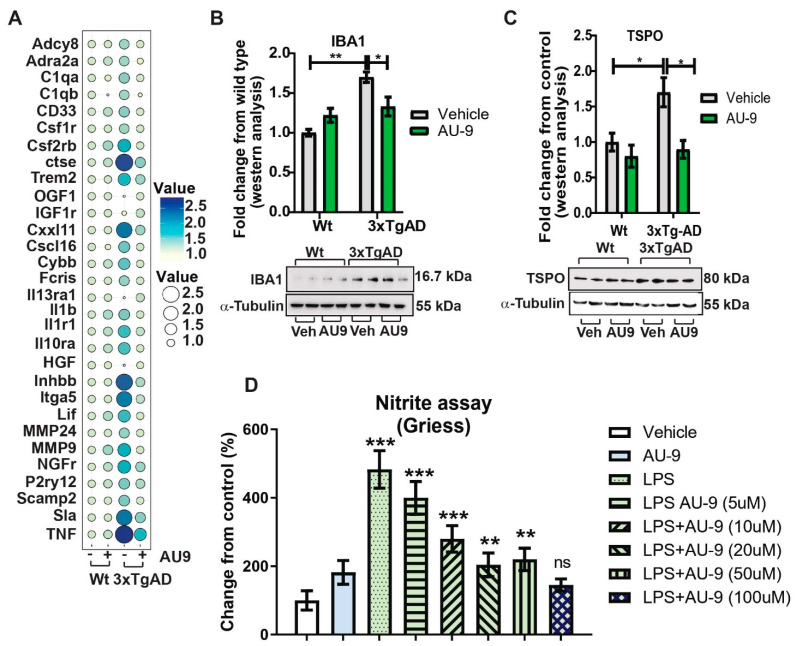
AU9 reduces inflammation. (**A**) Two-step qRTPCR analysis demonstrates by bubble plot that AU9 reduces inflammatory gene markers in 3xTgAD administered AU9 (5 mg/Kg daily for three months by oral gavage). (**B**,**C**) Western analysis demonstrates reduced protein expression of IBA1 and TSPO in similarly treated 3xTgAD mice as in A. Values were based on normalized protein concentrations, standardized to β-tubulin, and displayed as fold changes from control. Statistical values were obtained using Student *t*-test analysis ± S.E.M. Where *n* = 6 mice per group and *: *p* < 0.05 and **: *p* < 0.001. (**D**) Nitrite levels were measured by Griess reagent assay, where increasing concentrations of AU9 reduced LPS-mediated nitrite formation. Values were based upon triplicate readings from 6 independent assays and standardized to protein concentration. Statistical values were obtained using student *t*-test analysis ± S.E.M. *ns* = Not significant, **: *p* < 0.001, and ***: *p* < 0.0001.

**Figure 9 cells-12-01116-f009:**
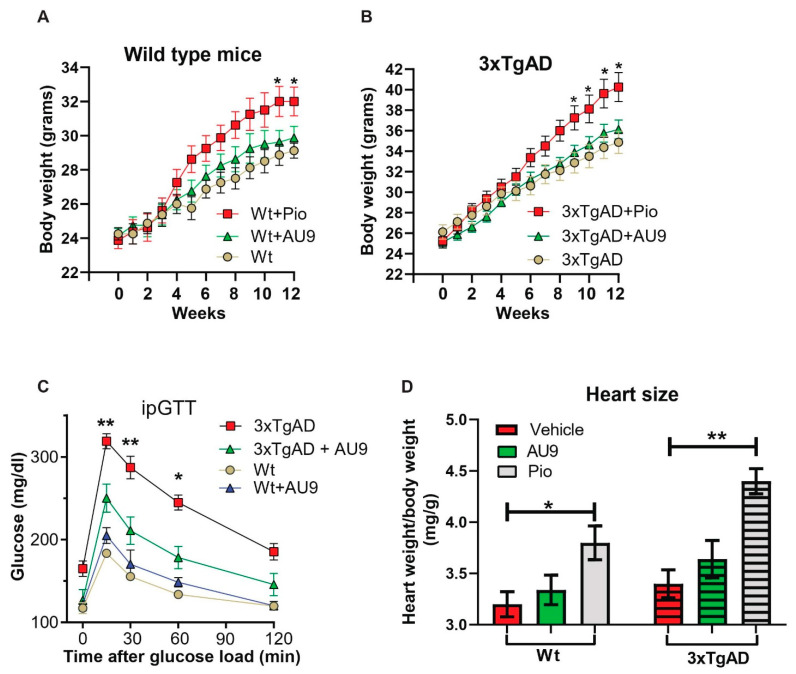
Physiological effects of AU9. (**A**,**B**) No significant weight change in 12-month-old wild type or 3xTgAD mice given AU9 (10 mg/Kg/day for 3 months, orally) compared to mice treated with Pio. (**C**) Intraperitoneal glucose tolerance test demonstrated that AU9 reduces circulating blood glucose in 3xTgAD mice. (**D**) Heart weight to body weight studies shows that AU9, when compared to Pio, does not induce an increase in size in age-matched wild-type and 3xTgAD mice. Statistical values were obtained using student *t*-test analysis ± S.E.M. Where *n* = 6 mice per group and *: *p* < 0.05 and **: *p* < 0.001.

## Data Availability

Not applicable.

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
