# Peer review of "Selective PPAR-Delta/PPAR-Gamma Activation Improves Cognition in a Model of Alzheimer’s Disease"

_cells, 2023, doi:10.3390/cells12081116_

Round 1
Reviewer 1 Report
Steinke and colleagues investigate the role of selective PPAR delta and gamma agonists improving AD (cognition, synaptic plasticity, and inflammation). The authors hypothesis that avoiding the Tyr-473 epitope in the PPAR gamma AF2 ligand binding domain will ultimate improve cognitive function associated with AD. It would be helpful for authors to include the rationalization for developing drugs that avoid the Tyr-473 epitope in the AF2 ligand binding domain. What evidence is there that this site was important/involved in the lack of benefit of the TZDs/PPAR agonists for improving behavioral deficits. Overall, the study design is sound and the experimental approach is adequate to address the question raised. The manuscript could be improved however with proper rationalization and the following:
1. There are several grammatical errors (misspelled words missing commas, etc). The authors should have someone to read the manuscripts who can provide grammatical edits. One example, is in the reporter assays section, “measurment” is spelled wrong Line 218.
2. The statistical test used is not clear. The methods state ANOVA with posthoc but also reveals that a t-test was used. It is not clear what test was used for data demonstrating statistical significance. For transparency, authors should include this information in the figure legend and/or results section so that it is clear. The appropriate test for most of the data would be ANOVA with posthoc. For some data a two-way ANOVA would be more appropriate. Perhaps the authors should consult with statistician to ensure proper analysis.
3. All bar graph data should also show individual points like Fig. 4.
4. It is not clear why some data was only in rosiglitazone and some in Pioglitazone. Authors should provide rationalization as to when one or the other was chosen.
Author Response
We would like to thank the reviewer for their kind comments and suggestions for improving our manuscript. Please find attached a point by point response to the comments in blue.

Reviewer 2 Report
This is a very well-executed study, and overall results are interpreted well and discussed in the context of previous findings. I have noted a few key points below that should be addressed, but overall these data are a significant and important contribution to the field. Please see my specific comments below:
Line 336 - 353: I would suggest that the authors move this information to a figure legend if they do not already feel the current figure legend is adequate as this is more descriptive about the figure presentation itself than actually describing the results.
Line 375, there is a typo in "duffer" and missing period.
Line 410-412: I would suggest the authors take care in making the statement that "AU9 is not as strong as an [typo] PPARdelta agonist as GW0742," unless there is a statistical difference between the two. Based on the data I would have concluded that they are similar, but at a minimum please do not make claims about differences that are not back by statistical comparison. If there is a statistical difference please make this more clear in the results section and appropriately note in figure.
line ~418: Please do not make claims about one result being "more significant" than another. The appropriate to make this comparison would be to do an ANOVA-type test with multiple comparisons corrected by a post-hoc test and then compare whether AU9 is significantly different from rosiglitazone and/or the CA-PPARg in the PPARgamma activation assay.
Line 421: Again, in line with my comments above, the term "weak" is rather unsupported in this context. Weaker than what? Is it statistically "weaker" than pioglitazone?
Is there a reason the authors switch to pioglitazone in the experiments for Fig 3E-G from Rosiglitazone in Fig 3D? I think either explaining the rationale or at least acknowledging that the results are with pioglitazone in the results section would be helpful, it's not even clear to me whether the "Pio" in the figure is a mistake since there's no mention of pioglitazone in the corresponding results. Indeed in line 424 results are discussed with respect to rosiglitazone?
Line 428: please provide citations for "activation of the PPARd and PPARg axis improves hippocampal cognition in mouse models for AD."
Line 438: suggest removing the sentence "AU9 treatment significantly improved the memory deficits in 3xTgAD mice." This is already reiterated at the end of the paragraph and it is more appropriate to summarize like this after presenting the results.
Line 449, I would argue that a p value of 0.88 is not a trend and the text "and that Au9 treated 3xTgAD mice trended towards and improvement," is grammatically incorrect and unclear. The authors do not need to try and make a stretch claim about the number of entries or worry about justifying it, it is what it is and doesn't take away from the result in my opinion. The more important result is (time spent in novel arm).
Line 450: Please do not make a claim about motor activity without the corresponding experiments to substantiate this. The Y-maze does not measure motor function.
Line 452: Please refrain from referring to "basal or baseline synaptic transmission" with respect to the input output curves. What you are really measuring is the ability of the field of neurons to translate a given amount of electrical stimulation of presynaptic fibers into a depolarization of the postsynaptic neuronal membranes. This is not the same thing as basal synaptic transmission, which you would want to measure using voltage clamp recordings of spontaneous EPSCs.
Line 456 and Fig 5A/B: Please note significance in the Figure panel for clarity. I would suggest increasing the range on the Y-axis so that differences can be more appreciated in Fig 5 panel A. Also please indicate how this significance testing was done, this cannot be done by just a one-way ANOVA as you note in the significance section of the methods.
Line 461: This result of reduced FV that is unable to be recovered suggests actual loss of presynaptic axonal input, not so much reduced "recruitment/activation." Would this be consistent with what we know about neuronal loss in these mice? Are there microscopic results showing loss of axonal density in this area in the literature that you cite?
Line 464: Based on the experiments in Fig 5 A-C there is not enough to conclude that you even observed reductions in basal synaptic transmission because you only record input-output responses (see my comment above). I would give the option to the authors because I do not think that the significance of this paper is necessarily diminished by not having these experiements; however, if they do wish to make a claim about the mechanism of action being related to postsynaptic signaling then I think a number of additional experiments would be required, including patch clamp experiments of the postsynaptic cells of interest. In addition, some indication that indeed "signaling" is actually affected would be required, as you cannot, based on the results presented in Fig 5, conclude that there are no off-target effects of the drug on glutamate receptors themselves for example.
Line 471: Again here, you are making claims that are too far beyond the data you are trying to interpret. AU9 ameliorates LTP deficits, beyond that you cannot say what is the mechanism without further investigation. Again I would leave it to the authors to decide if they wish to provide further experiments to support the claim that LTP deficits are rescued by modulating glutamatergic synaptic signaling. If further experiments do support this, then please wait to incorporate that conclusion until you have presented them in the current study, this would be more well received in the discussion for example.
Section "AU9 modulates glutamatergic receptor subunits in 3xTg-AD mice": Can the authors please comment on why they chose to look at the expression of post-synaptic proteins in a total protein fraction from tissue? I'm not sure that these data are interpretable because by doing this it simply implies that there are bulk changes in protein expression. First, this seems unlikely to have all these proteins change in overall expression, since especially the GluRs are typically regulated by changes in localization and/or phosphorylation. More importantly, the authors simply state they lysed the cells in "RIPA buffer." Please always indicate the exact components of any buffers used. In addition, the authors do not describe how they process these lysates, do you not centrifuge them to pellet debris before running them on the SDS-PAGE gel? Based on the ionic composition and detergent used in the lysis buffer, as well as the clarification of lysate step, you could be pelleting the compartment you are interested in looking at, and these changes could simply reflect changes in the localization to different compartments. I would ask that the authors reperform these experiments and do a very careful literature search on how to obtain different fractions of protein that correspond to different cellular compartments of interest. You can lyse the tissue in a non-detergent buffer and do a series of centrifugation steps while resuspending your pellets in different detergents to obtain fractions that have been shown to correspond to cellular compartments of interest. You can run all or most of these fractions to determine shifts of synaptic proteins between them, which would have more relevance to this study. Please see methods of Gustin et al., 2010. Tissue-specific variation of Ube3a protein expression in rodents and in a mouse model of Angelman syndrome. Neurobiol Dis, for an example of how this might be done, but there are other methods published that the authors could use to separate PSD-enriched fractions from membrane fractions and cytosolic fractions that could also be equally as informative. If the authors do not have frozen tissue available to complete these studies or mice that are already nearing the completion of treatment that could be used then I would not suggest that completely repeating these experiments is worth the cost and delay of publication to obtain that tissue. However, in this case, I would ask that the authors remove the data related to the total protein levels of synaptic proteins (Fig 5 F-H) as these experiments were performed with such confounding results (as noted above) that they could be misleading and take away from an otherwise outstanding study.
Author Response
We thank the reviewer for their many suggestions. We feel that the input form the reviewer will significantly help in making the current manuscript much better.

Reviewer 3 Report
General comments
The purpose of this paper is the development of new drugs for the treatment of Alzheimer's disease; a laudable cause. Unfortunately, several serious issues place the authors' conclusions at risk.
Behavioural paradigms do not appear to be well understood as the Y maze, as used, appears to be a forced alternation protocol but it was analysed as for spontaneous alternation.
Moreover, different group sizes appear to have been used for behavioural testing depending upon the outcome measure. Thus, behavioural results do not provide suitable evidence to support conclusions of a beneficial effect of the drug.
Group sizes are also an issue for in vitro work. For in vitro work, it is unclear whether data presented are mean±error of biologic or of technical replicates (although biologic replicates appear in the legends).
Neuropathology is also poorly described; for example, it is well-known that plaque density varies depending upon the specific area of cortex but no explanation on area analysed is provided.
Drug administration is also unclear – the methods suggest the drug was given by intraperitoneal injection but a figure legend suggests oral gavage.
Critically, the relevance of the data for translation is unclear.
Little information is provided on the expected distribution of AU9 – for example, it is dissolved in saline but it is expected to traverse into brain. Although the authors have previously published the generation of these compounds and have cited this paper, it is unclear from this paper how much of a water-soluble drug would enter brain normally. This important pharmacokinetic aspect has not been considered in the authors' discussion of shortcomings. This is of particular relevance to some in vivo findings – there appears to be a large variability in hippocampal amyloid deposition in the control group but no such variability is observed in the treatment group and thus, variability may contribute to this outcome measurement more than drug effect.
Similarly, the relevance of in vitro data using concentrations up to and including 100uM of the test drug to in vivo use in mice are unclear and are certainly unclear with respect to use of the drug in humans.
Even concentrations of 5uM and 10uM, used for many in vitro experiments, are very high for the in vivo state and to achieve in brain.
The most serious issue is the statistical analyses.
This is a very serious, major issue as the authors state they used 1-way ANOVAs, which are entirely inappropriate for the majority of data presented. T-tests with Welch's correction also appear to have been used, although it is unclear where it was used.
As a result of the poor statistics, the majority of conclusions are in question.
Specific issues
If Rosi and Pio do not cross the BBB, could the authors suggest why some results, cited in the introduction, are positive for treatment effect in mice and why some pilot trials in AD patients also showed some positive results?
Please clarify doses and treatment durations for in vivo work that were cited in citations 6-8 (line 60). Please list and refer to in vitro work separately to in vivo work.
Did the authors follow ARRIVE guidelines? If yes, please list how. If not, please justify.
E.g., Provide number of mice/cage
E.g., Clarify whether all mice within a particular cage received the same treatment.
E.g., Provide the number of mice treated with saline and with AU9, justified with sample size calculations and power analyses for predicted/expected/accepted treatment effects for all outcome measures use (behavioural, Western blotting, gene expression, Golgi staining, immunostaining etc).
E.g., clarify time of day for testing
Please provide information on the solubility/expected solubility of AU9.
Citation 11 (line 150) is not appropriate as a citation for a protocol for a behavioural test as the paper cited appears to be a review. Please use a more appropriate citation.
Please provide the baseline exploration of the objects during T1 (line 155).
Citation 12 (line 169) is not appropriate as a citation for a protocol for a behavioural test as the paper cited appears to be of a clinical trial in humans.
The Y maze task that was used was not spontaneous alternation – it appears to be of forced alternation. The novel arm would be expected to be explored as it has not been previously explored. It should therefore not be analysed as for spontaneous alternation but rather taking exploration of novelty into account.
What area of cortex was used for Western blotting (line 204)?
Provide the number of experiments (biologic replicates) for all in vitro experiments.
Provide information on whether data presented are from an individual replicate or are means calculated from biologic replicates.
What area of cortex was used for gene expression (line 234)?
How many neurons per section per mouse were analysed for Golgi staining?
What tissue was used for the β-secretase activity assay (line 264)?
What tissue source was used for the Aβ ELISA assay (line 268)?
Statistics: 1-way ANOVAs are not acceptable for comparison of data where there is more than one factor, e.g., Fig 3A contains two factors (dose (several levels) and treatment (two levels)), e.g., Fig 4A contains two factors (genotype and treament), e.g., Fig 5A and B each contain three factors (stimulus intensity (several levels), genotype (two levels) and treatment (two levels)). For example, data from qRT-PCR examine almost 30 transcripts (Fig 8) – a 1-way ANOVA is absolutely inappropriate. Repeated measures ANOVA should be used in several analyses.
Do the authors expect that concentrations of AU9 used for in vitro work (e.g., 10uM, line 729, or 5uM Fig 7C) are similar to concentrations expected in brain the in vivo experiments?
Figures: Please indicate number of technical replicates and of biologic replicates in all legends.
Figures: Please change to simple scatter plots so that all data points can be seen. Currently, the group sizes appear different in Fig 4A versus Fig 4B, 4C 4D 4E and 4F. Please also justify any difference in group size, should there actually be a difference in group size (Fig 4F versus 4A clearly show quite different groups sizes).
Figures: Fig 4F Time in novel arm is not the outcome explained in the Methods. A case for spontaneous alternation is difficult to justify based upon the Y maze protocol provided in the Methods. In addition, please justify the use of data from mice making very few entries – thus, failing an important confounding criterion of this task.
Results: 6E10 is well known to bind APP. Did the authors control for background staining for Fig 7B? The variability in the control group appears much larger than in the treated group. The authors are asked to comment on whether within-group variability played a role in outcome measure.
Method of administration to mice: compare line 793 with line 100.
Western blotting for IBA1 – the large variability within the TG AU9-treated group is not apparent in the relevant figure above (Fig 8B).
Figure 9D – groups are not apparent (colours appear very similar).
Unfortunately, there is consistent poor use of citations or poor checking of citations.
The meaning of lines 765 and of 804-805 are unclear (should refer to group size?).
Minor changes
Clarify legend for Fig 7 – it does not appear to follow the order shown in the graph.
Author Response
We wish to thank the reviewer for all their tremendous insight and frankly kind help in helping us make this a much better manuscript. We acknowledge countless errors in our initial submission. We hope we have clarified the countless mistakes we have made and hope the manuscript reads better.
In one point of clarity we have a extensive PK/PD study currently undergoing for an SBIR/STTR NIH application. We did not include the volumes of data we are getting from the study and hope to include this in a sperate manuscript along with the energy regulation and insulin signaling. We apologize for not including this data in the current manuscript and felt that it is important but that our current data is very incomplete and would take away from the new PK/PD study.

Reviewer 4 Report
Author clammed to discover PPAR- 34 delta- and PPAR-gamma agonist (AU9) that can ameliorate Alzheimer’s disease pathology without unwanted side effects. The manuscript is well designed and handles the situation related to PPAR activation and Alzheimer’s disease correlation comprehensively. On the other hand, there are some important points remains that should be addressed before the publication process.
1- There is no information related to possible systemic toxicity of AU9 compound. Is there possible organ accumulation of the compound that can affect haemostasis of the organ system like kidney, lung, and liver metabolism?
2- Similar approach should be exhibited with the healthy control (C57BL/6) and transgenic mice (3xTgAD).
3- Also, possible toxicological outcomes like genotoxicity related gene expression changes could be investigated.
4- AU9 treatment was shown to decrease Amyloid beta peptide accumulation. Is there possible effect of the compound that can alter gene expression of amyloid precursor protein (APP)?
5- There seemed to be a lot of information about the in silico approach in the article. Some of them are recommended to be shown in the supplementary material file.
Author Response
We thank the reviewer for their suggestions for improving the manuscript. Please find attached a point by point response to the concerns and suggestions.

Reviewer 5 Report
Alzheimer’s disease (AD) is a representative neurodegenerative disease with no effective drugs developed. Existing clinical drugs targeting PPARs show some extent of efficacy against AD, but also exhibit drawbacks, including safety issue and high dosing. In this work, Steinke and colleagues reported a dual PPAR-delta/PPAR-gamma agonist, AU9, designed first in silico. Following assessment in cells and animal model of AD showed effective activation of PPARs by AU9 and mitigation of AD phenotypes in 3xTgAD mouse model, including improvement of behavioral deficits, synaptic plasticity and reduction of amyloid-beta levels and inflammation. Noteworthy, AU9 may avoid the unwanted side effects of current PPAR-targeting drugs, even though this topic is not investigated in this work. Overall, this is an important study with developed AU9 compound showing a promising potential for further clinical investigation. The experiments are rational with well-designed controls and the data is generally supportive for the conclusion claimed by authors. However, the writing needs to be improved and the figure citation is little bit messy. I have listed some, but not all, of issues below. I strongly suggest authors to thoroughly go over the whole manuscript and make correction where appropriate.
line 32: remove "," between requiring and high dosing
line 66: Instead of "pioglitazone", the abbreviation Pio should be used here since it has been defined in the line 64.
Line 336: the figure citation here should be Fig. 1E-H instead of Fig. 2E-H.
The labeling in Fig. 1E-H and Fig. 2E-H is small and hard to recognize.
Line 410: The figure citation here "3A, and D" needs to be in a consistent form as in other places. Actually, Different citation forms have been used throughout the manuscript. The authors need to unify the format of all their figure citations.
Line 413: "interation" should be "interaction".
Line 418: It should be "Fig. 3F", not "Fig. 3". Also, in line 421, "Fig E" should be "Fig. 3E". I suggest authors to carefully securitize the figure citations.
Line 595: There are two "that", delete one.
Line 631: delete "," between "in the current" and "study we...".
Line 633: there are two "treated", delete one.
In the discussion part, it would be nice to use one paragraph to compare AU9 with other current PPAR agonists.
Author Response
We thank the reviewer for the kind suggestions. We have included a point by point response to the reviewer's concerns.

Round 2
Reviewer 2 Report
Thank you for your thoughtful responses to comments. I agree with all responses and/or actions taken by authors to address my comments. I would like to congratulate the authors on this important study and for the discovery and early development of this compound. I wish you all the best in all your future efforts! No further comments, and I recommend for publication of this updated manuscript.
Author Response
We thank the reviewer for all their kind help in making this a much better manuscript.
Reviewer 3 Report
I thank the authors for their attention to the many issues noted in my review.
I would now suggest a quick check for English (e.g., "However there exits evidence suggesting..." line 633 of amended manuscript - I think the authors mean "exists"; e.g., "These confounding results suggested that an laternative form of PPAR agonism..." line 641 of amended manuscript - I think the authors mean "alternative").
Author Response
I thank the authors for their attention to the many issues noted in my review.
I would now suggest a quick check for English (e.g., "However there exits evidence suggesting..." line 633 of amended manuscript - I think the authors mean "exists"; e.g., "These confounding results suggested that an laternative form of PPAR agonism..." line 641 of amended manuscript - I think the authors mean "alternative").
We thank the reviewer for their suggestions, we have corrected the suggested mistakes and reviewed the entire manuscript for any other additional mistakes. We thank the reviewer for all their comments and help in making our manuscript much better.
Reviewer 4 Report
The author did not perform a single modification related to arised concerns. In this format, I don't think the manuscript can be acceptable.
Author Response
We thank the reviewer for their suggestions. We have submitted our responses to the reviews in the first round of reviews.
